# One-Step Flow for Image Super-Resolution with Tunable Fidelity-Realism Trade-offs

**Yuanzhi Zhu**[*†1,2]  **Ruiqing Wang**[*1]  **Shilin Lu**[3]  **Junnan Li**[2]  **Hanshu Yan**[2]  **Kai Zhang**[‡1]
[1]Nanjing University    [2]Rhymes.AI    [3]Nanyang Technological University

## Abstract

Recent advances in diffusion and flow-based generative models have demonstrated remarkable success in image restoration tasks, achieving superior perceptual quality compared to traditional deep learning approaches. However, these methods either require numerous sampling steps to generate high-quality images, resulting in significant computational overhead, or rely on common model distillation, which usually imposes a fixed fidelity-realism trade-off and thus lacks flexibility. In this paper, we introduce OFTSR, a novel flow-based framework for one-step image super-resolution that can produce outputs with tunable levels of fidelity and realism. Our approach first trains a conditional flow-based super-resolution model to serve as a teacher model. We then distill this teacher model by applying a specialized constraint. Specifically, we force the predictions from our one-step student model for same input to lie on the same sampling ODE trajectory of the teacher model. This alignment ensures that the student model's single-step predictions from initial states match the teacher's predictions from a closer intermediate state. Through extensive experiments on datasets including FFHQ (256×256), DIV2K, ImageNet (256×256) and real world SR datasets, we demonstrate that OFTSR achieves state-of-the-art performance for one-step image super-resolution, while having the ability to flexibly tune the fidelity-realism trade-off. Code and pre-trained models are available at https://github.com/yuanzhizhu/OFTSR and https://huggingface.co/Yuanzhi/OFTSR, respectively.

## 1 Introduction

Recently, diffusion and flow-based generative models have demonstrated the ability to generate images with higher quality (Ramesh et al., 2022; Nichol & Dhariwal, 2021; Dhariwal & Nichol, 2021) than earlier generative models such as Generative Adversarial Networks (GANs) (Goodfellow et al., 2020; Karras et al., 2019), Normalizing Flows (NFs) (Dinh et al., 2016) and Variational Autoencoders (VAEs) (Kingma & Welling, 2013; Razavi et al., 2019). Beyond visual generation, diffusion models have shown remarkable success across a variety of tasks, including image editing (Hertz et al., 2022; Brooks et al., 2023; Kawar et al., 2023), 3D content generation (Poole et al., 2022; Wang et al., 2023a; Liu et al., 2023b; Wu et al., 2022; Wang et al., 2024a), and image restoration (Kawar et al., 2022; Chung et al., 2022; Wang et al., 2022b; Zhu et al., 2023; Delbracio & Milanfar, 2023; Lin et al., 2023), with particularly notable advancements in image super-resolution (SR) (Saharia et al., 2021; Chen et al., 2023; Yue et al., 2024b; Wang et al., 2024b).

Existing diffusion and flow-based SR methods can be broadly divided into two approaches: training-free methods (Zhu et al., 2023; Kawar et al., 2022; Wang et al., 2022b; Chung et al., 2022; Alkhouri et al., 2024; Mardani et al., 2023; Song et al., 2023a), and training-based methods (Saharia et al., 2021; Luo et al., 2023b; Liu et al., 2023a; Yue et al., 2023; Wang et al., 2024c; Yue et al., 2024b; Liu et al., 2024; Delbracio & Milanfar, 2023). Training-free methods decompose the conditional probability into a prior term and a likelihood term, with each term associating directly to a specific subproblem (Zhu et al., 2023). During iterative sampling, the prior subproblem is naturally handled by pre-trained unconditional diffusion models, which serve as powerful regularizers to guide

---

[*]Equal contribution.

[†]Work done while interned at Rhymes.AI (zyzeroer@gmail.com)

[‡]Corresponding author (kaizhang@nju.edu.cn)

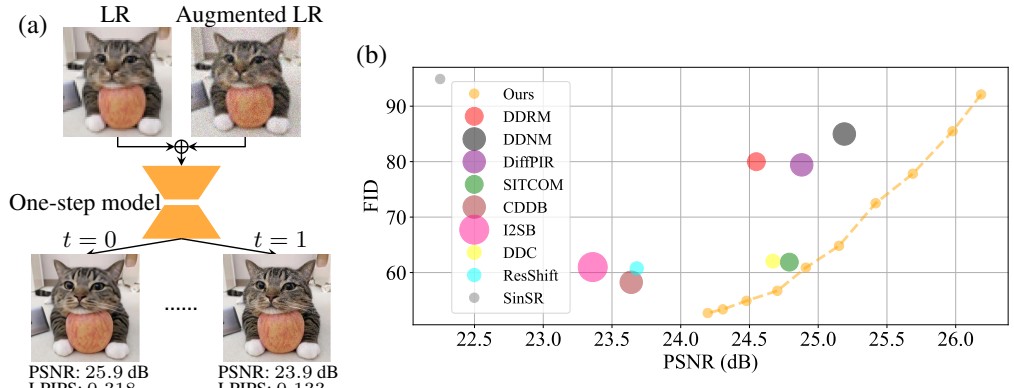

Figure 1: (a) Our final model takes the concatenation of a low-resolution image with its noise-augmented version as input, and is able to generate high-resolution outputs with either high realism or high fidelity by adjusting the interpolation parameter $t$. We indicate the PSNR and LPIPS value on the output images. (b) Comparison of different diffusion and flow based image super-resolution methods on the ImageNet $256 \times 256$ dataset. Bubble radius indicates the NFEs used by the methods.

the solution toward realistic High Resolution (HR) images. Meanwhile, the likelihood subproblem is addressed through specialized optimization techniques or analytical approximations to ensure fidelity to the observed Low Resolution (LR) image. On the other hand, training-based methods directly model the conditional probability using paired data, either by training from scratch (Saharia et al., 2021; Delbracio & Milanfar, 2023) or by incorporating additional control modules into existing generative priors (Wang et al., 2024b; Yu et al., 2024; Lin et al., 2023; Rombach et al., 2022). Several other bridge-based methods (Luo et al., 2023b; Liu et al., 2023a; Yue et al., 2023; Chung et al., 2024) have also been proposed for general image-to-image translation tasks, sharing similarities with direct learning approaches.

Despite the promising results of the above methods, they require many iterative sampling steps to achieve high perceptual quality, and reducing the number of iterations often results in higher fidelity but lower perceptual quality. In this sense, their fidelity-realism trade-offs are achieved at the cost of more sampling steps. In order to achieve high perceptual quality with fewer sampling steps, some attempts (Wang et al., 2024c; Lee et al., 2024; Wu et al., 2024; Xie et al., 2024; Li et al., 2024) have been made to distill the diffusion sampling process into a single step with diffusion distillation approaches (Luhman & Luhman, 2021; Salimans & Ho, 2022; Liu et al., 2022; Song et al., 2023b; Yan et al., 2024; Yin et al., 2024b;a; Sauer et al., 2025). However, while these methods improve efficiency, they sacrifice flexibility by limiting control over the fidelity-realism trade-off, reducing their applicability in domains where different tasks require varying levels of fidelity and realism, such as medical imaging, remote sensing and film upscaling (Greenspan, 2009; Li et al., 2023a; Wang et al., 2022a; Mentzer et al., 2020; Joshi et al., 2025).

In this paper, we propose OFTSR that achieves one-step image SR and preserves the capability to produce outputs with tunable fidelity-realism trade-offs. Specifically, OFTSR uses a two-stage pipeline. In stage one we train a noise-augmented conditional rectified flow to expand the support of the initial distribution: noise-perturbed LR images form the initial distribution while the LR images are used as conditions, enabling diverse HR reconstructions from a single LR. In the second stage, a distillation strategy is proposed to restrict the student model's predictions to match the same Ordinary Differential Equation (ODE) induced by the teacher model from the first stage.

Our main contributions can be summarized as follows:

- **Noise-augmented Conditional Rectified Flow for Image Restoration**: We introduce an enhanced conditional rectified flow model for image restoration. By leveraging a noise-augmented LR conditioning strategy, our approach enables more effective LR-conditioned diffusion restoration, serving as both a general restoration framework and the foundational stage for our proposed distillation algorithm.

- **One-Step Diffusion Distillation with Flexible Fidelity-Realism Trade-off**: We introduce a distillation strategy applicable to empirical probability flow ODEs of *any* pre-trained conditional diffusion or flow model. Unlike prior methods that limit flexibility, ours enables one-step sampling while preserving control over fidelity and perceptual realism for SR.

- **State-of-the-Art (SOTA) Performance on Benchmark Datasets**: Extensive experiments on DIV2K (Agustsson & Timofte, 2017), FFHQ (Karras et al., 2019), ImageNet (Deng et al., 2009) and several real world SR dataset including RealSR (Cai et al., 2019), RealSet80 (Yue et al., 2024b) and RealLQ250 (Ai et al., 2025) show that OFTSR achieves competitive one-step reconstruction, surpassing recent SOTA methods in both perceptual quality and fidelity.

## 2 BACKGROUND

### 2.1 DIFFUSION AND FLOW-BASED GENERATIVE MODELS

Drawing inspiration from non-equilibrium thermodynamics, diffusion models operate through two core processes: a forward diffusion process that gradually adds Gaussian noise to data until it becomes pure noise, and a reverse denoising process that systematically reconstructs the original data by removing noise (Sohl-Dickstein et al., 2015; Ho et al., 2020; Song et al., 2020b). Let $\mathbf{x}_t$ represent the data $\mathbf{x}$ at timestep $t$. The forward process can be formally described by the Itô Stochastic Differential Equation (SDE) (Song et al., 2020b):

$$\mathrm{d}\mathbf{x}_t = f_t \mathbf{x}_t \mathrm{d}t + g_t \mathrm{d}\mathbf{w}, \tag{1}$$

where $\mathbf{w}$ is the standard Wiener process, $f_t : \mathbb{R} \to \mathbb{R}$ is the drift coefficient, and $g_t : \mathbb{R} \to \mathbb{R}$ is a scalar function called the diffusion coefficient.

For every diffusion process described by Eq. (1), there exists a corresponding deterministic Probability Flow Ordinary Differential Equation (PF-ODE) that maintains the same marginal probability density:

$$\frac{\mathrm{d}\mathbf{x}_t}{\mathrm{d}t} = f_t \mathbf{x}_t - \frac{1}{2} g_t^2 \nabla_{\mathbf{x}_t} \log p_t(\mathbf{x}_t), \tag{2}$$

where $p_t(\cdot)$ represents the marginal probability density at time $t$. The term $\nabla_{\mathbf{x}_t} \log p_t(\mathbf{x}_t)$ is known as the score function, which can be approximated by a neural network $\mathbf{s}_\theta(\mathbf{x}, t)$ with parameters $\theta$. This network is typically trained using score matching techniques (Hyvärinen & Dayan, 2005; Song & Ermon, 2019; Song et al., 2020a).

To generate data samples, the process begins with Gaussian noise drawn from an initial Gaussian distribution $p_0$ and solves Eq. (2) numerically from $t = 0$ to $t = 1$. By utilizing the learned score function $\mathbf{s}_\theta(\mathbf{x}_t, t)$, the empirical PF-ODE can be obtained as: $\frac{\mathrm{d}\mathbf{x}_t}{\mathrm{d}t} = f_t \mathbf{x}_t - \frac{1}{2} g_t^2 \mathbf{s}_\theta(\mathbf{x}_t, t)$.

Rectified flow (Liu et al., 2022; Liu, 2022; Lipman et al., 2022; Esser et al., 2024) is a generative modeling framework based on ODEs. Given an initial distribution $p_0$ and a target data distribution $p_1$, rectified flow trains a neural network to parameterize a velocity field using the following loss function:

$$\mathcal{L}_{\mathrm{rf}}(\theta) := \mathbb{E}_{\mathbf{x}_1 \sim p_1, \mathbf{x}_0 \sim p_0} \left[ \int_0^1 \left\| \mathbf{v}_\theta(\mathbf{x}_t, t) - (\mathbf{x}_1 - \mathbf{x}_0) \right\|_2^2 \mathrm{d}t \right], \text{ where } \mathbf{x}_t = (1-t)\mathbf{x}_0 + t\mathbf{x}_1. \tag{3}$$

Once trained, sample generation is achieved by solving the empirical ODE $\frac{\mathrm{d}\mathbf{x}_t}{\mathrm{d}t} = \mathbf{v}_\theta(\mathbf{x}_t, t)$ from $t = 0$ to $t = 1$. In practical implementations, this empirical ODE is solved numerically using standard ODE solvers, ranging from the simple forward Euler method to higher-order methods such as RK2 and RK45.

### 2.2 PERCEPTION-DISTORTION TRADE-OFF

The perception-distortion (realism-fidelity) trade-off (Blau & Michaeli, 2018) is a fundamental concept in image restoration. It describes the inherent trade-off between perceptual realism and fidelity to the ground truth, and mathematically proves that it is generally not possible to achieve both good perceptual realism and high fidelity simultaneously.

To address this challenge, researchers have explored various approaches to enable tunable trade-offs between these two desirable qualities. One common technique involves interpolating between the weights of two models with the same architecture, trained with GAN loss and mean squared error loss (Wang et al., 2018). Recently, diffusion models have emerged as a promising approach for

this task. The iterative sampling nature of diffusion models provides a flexible means of controlling the desired trade-offs. By adjusting the Number of Function Evaluations (NFEs), users can generate reconstructions that better match their specific requirements (Chung et al., 2024). Specifically, lower NFEs tend to result in reconstructions with reduced distortion, as the output regresses towards the mean (Delbracio & Milanfar, 2023). Conversely, higher NFEs prioritize perceptual quality, even if it comes at the expense of some distortion from the ground truth (similar to Fig. 6).

## 3 METHOD

In this section, we introduce the OFTSR framework for one-step SR that can restore HR images with either high realism or high fidelity. We achieve this goal through a two-stage process: first, we train a direct flow-based model for SR, and then we distill this model into a simplified one-step variant. In Sec. 3.1, we present a simple noise-augmented conditional flow that expands the support of the initial distribution, enabling diverse reconstruction. In Sec. 3.2, we propose to distill the student model by restricting its predictions on the same ODE using teacher model from Sec. 3.1.

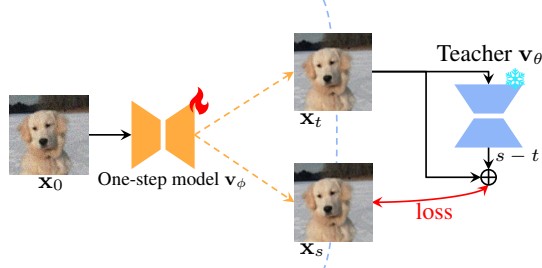

Figure 2: Illustration of the proposed distillation loss. Rather than directly distilling from the teacher, we leverage the teacher to align the one-step intermediate outputs, $\mathbf{x}_t$ and $\mathbf{x}_s$, along teacher's PF-ODE trajectory. For simplicity, LR conditioning is omitted in this figure.

### 3.1 NOISE AUGMENTED CONDITIONAL FLOW

Unlike diffusion models, flow-based models have the advantage that their initial distribution is not limited to Gaussian distributions. This flexibility suggests a natural approach for image restoration - directly learning a flow that maps the distribution of LR images ($p_{\text{LR}}$) to that of HR images ($p_{\text{HR}}$). However, our initial experiments (see Tab. 7) showed poor performance with this direct approach, aligning with findings from several recent works (Delbracio & Milanfar, 2023; Kim et al., 2024; Lee et al., 2024). This training procedure tends to collapse the LR→HR mapping: during inference each LR is driven toward a single HR.

To overcome this limitation, we propose a noise-augmented approach to process LR images. For any input image $\mathbf{x}_{\text{LR}}$, we construct our initial distribution $p_0(\mathbf{x}) = p_{\text{LR}}^{\sigma_p}$ by adding Gaussian noise with standard deviation $\sigma_p$. Specifically, we adopt a Variance-Preserving (VP) noising operation (Ho et al., 2020; Song et al., 2020b):

$$\mathbf{x}_0 = \sqrt{1 - \sigma_p^2}\mathbf{x}_{\text{LR}} + \sigma_p\epsilon, \tag{4}$$

where $\epsilon$ is a standard Gaussian noise. While this noise perturbation facilitates better generalization, it inevitably causes information loss in the LR image. To address this, we incorporate $\mathbf{x}_{\text{LR}}$ as a conditional input to our model as in Fig. 1. This VP formulation, together with the condition $\mathbf{x}_{\text{LR}}$, makes our method particularly versatile, encompassing previous approaches as special cases. When $\sigma_p = 0$, our method reduces to the minimal augmentation case in InDI (Delbracio & Milanfar, 2023), and when $\sigma_p = 1$, it matches the training strategy of SR3 (Saharia et al., 2022).

Given this noise-augmented formulation, we can now define our training objective as:

$$\mathcal{L}_{\text{flow}}(\theta) = \mathbb{E}_{\mathbf{x}_1 \sim p_1}\left[\int_0^1 \mathbb{D}\bigg(\mathbf{v}_\theta(\mathbf{x}_{t,\text{LR}}, t), (\mathbf{x}_1 - \mathbf{x}_0)\bigg)dt\right], \tag{5}$$

where $\mathbb{D}$ is a discrepancy loss that measures the difference between two images (e.g., $\ell_2$ loss or the $\ell_1$ loss), $\mathbf{v}_\theta$ is our velocity model, $\mathbf{x}_{t,\text{LR}} = \text{concat}(\mathbf{x}_t, \mathbf{x}_{\text{LR}})$ is the concatenation of $\mathbf{x}_t$ and $\mathbf{x}_{\text{LR}}$ in channel dimension (see Fig. 1), The LR input of the algorithm is given by $\mathbf{x}_{\text{LR}} = \mathcal{H}^T(\mathcal{H}(\mathbf{x}_1) + \mathbf{n})$, where $\mathcal{H}$ is the downsampling operator, $\mathcal{H}^T$ is its transpose and $\mathbf{n}$ is i.i.d. Gaussian noise with variance $\sigma_n^2$. The perturbed version of $\mathbf{x}_{\text{LR}}$, denoted as $\mathbf{x}_0$, is obtained using the noise augmentation strategy described in Eq. (4). Additionally, $\mathbf{x}_t = (1 - t)\mathbf{x}_0 + t\mathbf{x}_1$ denotes the intermediate state as in rectified flow (Liu et al., 2022; Liu, 2022).

## 3.2 DISTILLATION LOSS

We introduce a distillation loss to train a one-step student that preserves the pre-trained SR flow's fidelity–realism trade-off, allowing control at inference via a single hyperparameter $t$. As shown in Fig. 6 and observed in prior work (Delbracio & Milanfar, 2023; Liu et al., 2023a), single-step estimates of the *final state* $\mathbf{x}_1^t$ obtained from an *intermediate state* $\mathbf{x}_t$ lie on a fidelity–realism curve: along the ODE sampling trajectory, estimates for larger $t$ (closer to 1) exhibit richer detail and lower LPIPS (better realism), whereas estimates for smaller $t$ (closer to 0) are blurrier but achieve lower MMSE and higher PSNR (better fidelity).

To preserve the fidelity-realism trade-off, given the same input $\mathbf{x}_{0,\mathrm{LR}}$, for two different timesteps $t$ and $s$ where $s > t$, we require the student model $\mathbf{v}_\phi$ to produce two corresponding intermediate states $\mathbf{x}_t$ and $\mathbf{x}_s$ that lie on the same ODE trajectory defined by the teacher (see Fig. 2):

$$\mathbf{x}_s = \mathbf{x}_t + (s - t)\mathbf{v}_\theta(\mathbf{x}_{t,\mathrm{LR}}, t), \tag{6}$$

where $\mathbf{x}_{0,\mathrm{LR}} = \mathrm{concat}(\mathbf{x}_0, \mathbf{x}_{\mathrm{LR}})$ is the concatenation of the input image $\mathbf{x}_0$ and the LR condition $\mathbf{x}_{\mathrm{LR}}$ along the channel dimension. The intermediate states $\mathbf{x}_t$ and $\mathbf{x}_s$ can be computed using our one-step student model $\mathbf{v}_\phi$:

$$\mathbf{x}_t = \mathbf{x}_0 + t\mathbf{v}_\phi(\mathbf{x}_{0,\mathrm{LR}}, t). \tag{7}$$

Substituting the expression for the intermediate image $\mathbf{x}_t$ and $\mathbf{x}_s$ from Eq. (7) into Eq. (6), we have the following constraint on the student model:

$$s(\mathbf{v}_\phi(\mathbf{x}_{0,\mathrm{LR}}, s) - \mathbf{v}_\phi(\mathbf{x}_{0,\mathrm{LR}}, t)) = (s - t)(\mathbf{v}_\theta(\mathbf{x}_{t,\mathrm{LR}}, t) - \mathbf{v}_\phi(\mathbf{x}_{0,\mathrm{LR}}, t)). \tag{8}$$

Similar to BOOT, we can set $\mathrm{d}t = s - t$ and derive the final distillation loss:

$$\mathcal{L}_{\mathrm{distill}}(\phi) = \mathbb{E}_{\mathbf{x}_1 \sim p_1, t \sim \mathcal{U}[0,1]} \left[ \left\| \mathbf{v}_\phi(\mathbf{x}_{0,\mathrm{LR}}, s) - \mathrm{SG}\left[ \mathbf{v}_\phi(\mathbf{x}_{0,\mathrm{LR}}, t) + \frac{\mathrm{d}t}{s}\left( \mathbf{v}_\theta(\mathbf{x}_{t,\mathrm{LR}}, t) - \mathbf{v}_\phi(\mathbf{x}_{0,\mathrm{LR}}, t) \right) \right] \right\|_2^2 \right], \tag{9}$$

where $\mathrm{SG}[\cdot]$ is the stop-gradient operator for training stability (Gu et al., 2023; Tee et al., 2024). Since $s - t = \mathrm{d}t$ and $t > 0$, we do not have the 'dividing by 0' issue in (Tee et al., 2024). Similarly to (Song et al., 2023b; Gu et al., 2023), we can use the Euler or general RK2 solver to calculate $\mathbf{v}_\theta$ in Eq. (9). In our main experiments, we employ the midpoint method, while also evaluating two other RK2 solver variants, *i.e.*, Heun's method and Ralston's method, for comparison in our ablations (see Tab. 8). In Sec. B.2, we show that our distillation loss is the discrete-time counterpart of the forward distillation loss (Boffi et al., 2025; Liu, 2025) by fixing the start timestep at 0, which is highly related to recent work MeanFlow (Geng et al., 2025) and AlignYourFlow (Sabour et al., 2025).

## 3.3 ALIGNMENT AND BOUNDARY LOSS

In BOOT (Gu et al., 2023), a boundary condition is applied to enforce that the one-step student model and teacher model perform the same at the boundary $t = 0$. We aim to align the teacher and student outputs in our model. The student produces $\mathbf{x}_0 + \mathbf{v}_\phi(\mathbf{x}_{0,\mathrm{LR}}, t)$, while the teacher generates $\mathbf{x}_t + (1-t)\mathbf{v}_\theta(\mathbf{x}_{t,\mathrm{LR}}, t)$ based on the student's output $\mathbf{x}_t$ using Eq. (7). By minimizing the difference between these outputs, we get the following alignment loss to align the teacher and student:

$$\mathcal{L}_{\mathrm{align}}(\phi) = \mathbb{E}_{\mathbf{x}_1 \sim p_1, t \sim \mathcal{U}[0,1]} \left[ \left\| (1 - t)\left( \mathbf{v}_\phi(\mathbf{x}_{0,\mathrm{LR}}, t) - \mathbf{v}_\theta(\mathbf{x}_{t,\mathrm{LR}}, t) \right) \right\|_2^2 \right]. \tag{10}$$

If we consider this alignment loss only at $t = 0$, it becomes equivalent to the boundary loss used in BOOT:

$$\mathcal{L}_{\mathrm{BC}}(\phi) = \mathbb{E}_{\mathbf{x}_1 \sim p_1} \left[ \left\| \mathbf{v}_\phi(\mathbf{x}_{0,\mathrm{LR}}, 0) - \mathbf{v}_\theta(\mathbf{x}_{0,\mathrm{LR}}, 0) \right\|_2^2 \right]. \tag{11}$$

Since it is difficult to sample $t = 0$ for most training iterations, we add in addition the boundary loss Eq. (11) in our final training objective.

**The overall training objective.** The student network $\mathbf{v}_\phi$ is trained to minimize the combination of the aforementioned three losses terms:

$$\mathcal{L}(\phi) = \mathcal{L}_{\mathrm{distill}}(\phi) + \lambda_{\mathrm{align}}\mathcal{L}_{\mathrm{align}}(\phi) + \lambda_{\mathrm{BC}}\mathcal{L}_{\mathrm{BC}}(\phi), \tag{12}$$

where $\lambda_{\mathrm{align}}$ and $\lambda_{\mathrm{BC}}$ are the weights for alignment loss and boundary condition loss, respectively. The distillation stage of the proposed method is summarized in Algorithm 1.

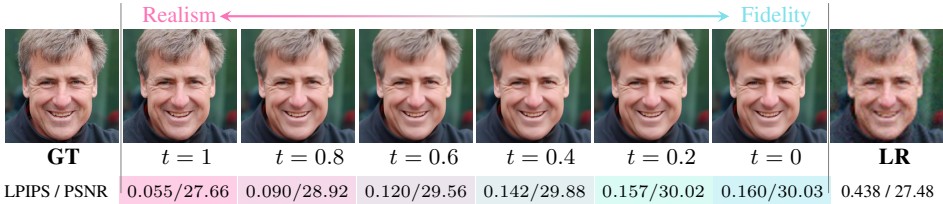

Figure 3: OFTSR is capable to generate continuous transitions between image realism and fidelity.

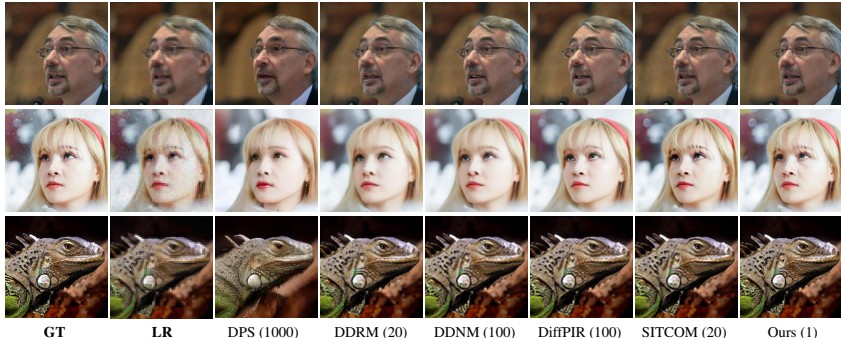

Figure 4: **Qualitative comparison with training-free methods.** The first row shows noiseless SR on the FFHQ dataset, the second row presents noisy SR ($\sigma_n = 0.05$) on FFHQ, and the bottom row demonstrates noiseless SR on the ImageNet dataset. Numbers next to the method names represent the required NFEs.

**Inference.** After training, the one-step student $\mathbf{v}_\phi$ produces the final high-resolution output $\mathbf{x}_1^t$ in a single forward pass, conditioned on the initial state $\mathbf{x}_0$, the low-resolution input $\mathbf{x}_{\text{LR}}$, and the trade-off parameter $t$. Concretely,

$$\mathbf{x}_1^t = \mathbf{x}_0 + \mathbf{v}_\phi(\mathbf{x}_0, \mathbf{x}_{\text{LR}}, t), \tag{13}$$

where $\mathbf{v}_\phi(\cdot)$ predicts a residual that refines $\mathbf{x}_0$ toward the desired point on the fidelity–realism curve specified by $t$.

### 3.4 COMPARISON TO RELATED WORKS

In this section, we distinguish the proposed OFTSR from several closely related methods.

**BOOT (Gu et al., 2023).** Gu *et al.* proposed to make the prediction of the student model fulfill the Signal-ODE. In contrast, OFTSR directly constrains the student's implicit prediction $\mathbf{x}_t$ using the PF-ODE of the teacher model, leading to more concise and intuitive derivation and distillation objective. Moreover, while BOOT was originally designed for text-to-image generation using diffusion models, our method is built on rectified flow and demonstrates a smaller distillation gap compared to BOOT loss for SR task, and empirically achieves markedly better fidelity–realism trade-offs.

**DAVI (Lee et al., 2024).** Lee *et al.* introduced DAVI, which combines Variational Score Distillation (VSD) loss (Wang et al., 2024d; Luo et al., 2023a; Yin et al., 2024b) with data consistency loss to train a one-step SR model and utilizes the perturbation trick to present robust restoration ability. However, DAVI needs to train a fake score to track the denoising score of the one-step generator, resulting in reduced training efficiency.

**SinSR (Wang et al., 2024c).** Wang *et al.* proposed SinSR, which achieves near-teacher performance by distilling ResShift (Yue et al., 2024b) without adversarial training. However, SinSR requires simulation of the teacher model's ODE trajectory, leading to computational overhead during training.

Our OFTSR stands out from other diffusion and flow-based SR methods due to its unique ability to restore images with either high perceptual quality or low distortion. This capability is novel among diffusion and flow-based approaches.

## 4 EXPERIMENTS

In this section, we provide experimental details and empirical evaluation of OFTSR and compare it with prior works.

Table 1: Noiseless quantitative results on **DIV2K**. We compute the average PSNR (dB), LPIPS and FID of different methods on 4× SR. The best and second best results are highlighted in **bold** and underline. The distilled model produces superior performance in terms of trade-off metrics through adjustment of the hyperparameter $t$.

| DIV2K | Method | NFEs (↓) | PSNR (↑) | LPIPS (↓) | FID (↓) |
|---|---|---|---|---|---|
| Training-free | DPS (Chung et al., 2022) | 1000 | 23.05 | 0.447 | 109.35 |
| | DDRM (Kawar et al., 2022) | 20 | 27.87 | 0.285 | 23.38 |
| | DDNM (Wang et al., 2022b) | 100 | 28.09 | 0.279 | 20.33 |
| | DiffPIR (Zhu et al., 2023) | 100 | 27.94 | 0.248 | 19.56 |
| Training-based | IRSDE (Luo et al., 2023b) | 100 | 26.83 | 0.144 | 14.69 |
| | GOUB (Yue et al., 2023) | 100 | 26.92 | 0.218 | 21.56 |
| | ECDB (Yue et al., 2024a) | 100 | 27.39 | 0.212 | 18.88 |
| | InDI (Delbracio & Milanfar, 2023) | 100 | 26.45 | 0.136 | 15.39 |
| | Ours | 31 | 26.76 | 0.128 | **14.10** |
| | Ours distilled ($t = 1$) | 1 | 26.87 | **0.127** | 14.58 |
| | Ours distilled ($t = 0.5$) | 1 | 28.02 | 0.208 | 16.89 |
| | Ours distilled ($t = 0$) | 1 | **28.99** | 0.271 | 18.07 |

Table 2: Noiseless (top) and noisy (bottom) quantitative results on **FFHQ 256×256**. We compute the average PSNR (dB), LPIPS and FID of different methods on 4× SR. The best and second best results are highlighted in **bold** and underline.

| FFHQ | | Method | NFEs (↓) | PSNR (↑) | LPIPS (↓) | FID (↓) |
|---|---|---|---|---|---|---|
| $\sigma_n = 0$ | Training-free | DPS (Chung et al., 2022) | 1000 | 24.08 | 0.180 | 79.71 |
| | | DDRM (Kawar et al., 2022) | 20 | 28.81 | 0.118 | 89.12 |
| | | DDNM (Wang et al., 2022b) | 100 | 29.45 | 0.091 | 60.99 |
| | | DiffPIR (Zhu et al., 2023) | 100 | 29.13 | 0.073 | 44.49 |
| | | SITCOM (Alkhouri et al., 2024) | 20 | 29.29 | 0.089 | 43.00 |
| | Training-based | Ours | 20 | 28.83 | **0.053** | **30.54** |
| | | Ours distilled ($t = 1$) | 1 | 28.98 | 0.055 | 36.02 |
| | | Ours distilled ($t = 0.5$) | 1 | 29.95 | 0.093 | 49.08 |
| | | Ours distilled ($t = 0$) | 1 | **31.25** | 0.150 | 66.76 |
| $\sigma_n = 0.05$ | Training-free | DPS (Chung et al., 2022) | 1000 | 23.61 | 0.186 | 81.25 |
| | | DDRM (Kawar et al., 2022) | 20 | 26.71 | 0.191 | 113.25 |
| | | DDNM (Wang et al., 2022b) | 100 | 27.66 | 0.174 | 113.26 |
| | | DiffPIR (Zhu et al., 2023) | 100 | 26.99 | 0.123 | 61.66 |
| | | SITCOM (Alkhouri et al., 2024) | 20 | 27.80 | 0.158 | 83.04 |
| | Training-based | DAVI (Lee et al., 2024) | 1 | 27.50 | 0.084 | 50.19 |
| | | Ours | 20 | 27.28 | **0.080** | **46.04** |
| | | Ours distilled ($t = 1$) | 1 | 27.71 | 0.081 | 49.81 |
| | | Ours distilled ($t = 0.5$) | 1 | 29.47 | 0.157 | 82.93 |
| | | Ours distilled ($t = 0$) | 1 | **29.75** | 0.172 | 85.89 |

## 4.1 EXPERIMENTAL SETUP

**Datasets.** We perform extensive SR experiments on the FFHQ 256×256 (Karras et al., 2019), DIV2K (Agustsson & Timofte, 2017) and ImageNet 256×256 (Russakovsky et al., 2015) datasets to assess the bicubic SR performance of OFTSR on faces and natural images. For each dataset, we evaluate on 100 hold-out validation images without cherry-picking. For evaluating real SR, we use both synthetic set and real world set. Synthetic set includes 100 images from Imagenet for 64→256 SR and 100 images from DIV2K for 128→512 SR, both degraded using RealESRGAN pipeline. Real world test set includes RealSR (Cai et al., 2019), RealSet80 (Yue et al., 2024b) and RealLQ250 (Ai et al., 2025). For distilling DiT4SR, we construct the training set using a combination of images from DIV2K (Agustsson & Timofte, 2017), DIV8K (Gu et al., 2019), Flickr2K (Timofte et al., 2017), LSDIR (Li et al., 2023b) and the first 10K images from FFHQ (Karras et al., 2019).

**Teacher Models.** We employ three types of teacher models in our experiments: (1) Self-trained teachers (*2 types of backbones*: Guided Diffusion (Dhariwal & Nichol, 2021) for bicubic SR (Tabs. 1 to 3) and ResShift (Yue et al., 2024b) for real SR (Tabs. 4 and 5)) using the noise-augmented conditional flow strategy in Sec. 3.1, which showcases the effectiveness of our training scheme; (2) An off-the-shelf DiT4SR teacher (built on Stable Diffusion (SD) 3.5). Since SD-based models possess significantly stronger generative priors and are prohibitively expensive for us to pre-train, we distill DiT4SR with our method to enable fair comparison with the latest SOTA approaches for realSR (Tab. 6); (3) An off-the-shelf ResShift teacher, allowing direct comparison with SinSR (which is distilled from ResShift) and fair computational cost comparison (Tabs. 9 and 10).

**Evaluation Metrics.** The metrics we use for comparison are Peak Signal-to-Noise Ratio (PSNR), Fréchet Inception Distance (FID), and Learned Perceptual Image Patch Similarity (LPIPS) (Zhang et al., 2018) distance. The FID evaluates the visual quality by calculating the feature distance between two image distributions. In our experiments, we calculate the FID using the HR images and the restored images from the 100 hold-out validation set with Clean-FID (Parmar et al., 2022). LPIPS measures the average perceptual similarity between the restored images and their corresponding HR images. PSNR measures the restoration faithfulness between two images. And LPIPS and PSNR are the two main metrics we use to measure the perceptual-fidelity trade-offs. For real SR task, we also use no-reference Image Quality Assessment (IQA) including NIQE Zhang et al. (2015), CLIPIQA Wang et al. (2023b), MUSIQ Ke et al. (2021), and MANIQA Yang et al. (2022).

**Compared Methods.** We conduct comprehensive comparisons against state-of-the-art diffusion-based image super-resolution methods, which can be categorized into two groups: (1) Training-free methods, including DPS (Chung et al., 2022), DDRM (Kawar et al., 2022), DDNM (Wang et al., 2022b), DiffPIR (Zhu et al., 2023), CDDB (Chung et al., 2024), and SITCOM (Alkhouri et al., 2024); (2) Training-based methods: GOUB (Yue et al., 2023), ECDB (Yue et al., 2024a), InDI (Delbracio & Milanfar, 2023), DAVI (Lee et al., 2024), I2SB (Liu et al., 2023a), DDC (Chen et al., 2024), ResShift (Yue et al., 2024b), SinSR (Wang et al., 2024c) and CTMSR (You et al., 2025). And large scale Stable Diffusion based methods such as OSEDiff (Wu et al., 2024), AddSR (Xie et al., 2024) and TSDSR (Dong et al., 2025). It is noteworthy that SITCOM requires K inner-iterations to evaluate and differentiate the score function at each sampling step. To further validate the effectiveness of our method, we conduct experiment on real-world image super-resolution. Following (Yue et al., 2024b; Wang et al., 2024c), we use Imagenet 256 × 256 as HR training data and synthesize LR images using degradation pipeline of RealESRGAN (Wang et al., 2021).

Table 3: Noiseless quantitative results on ImageNet 256×256. We compute the average PSNR (dB), LPIPS and FID of different methods on 4× SR. The best and second best results are highlighted in **bold** and underline.

| ImageNet | Method | NFEs (↓) | PSNR (↑) | LPIPS (↓) | FID (↓) |
|---|---|---|---|---|---|
| Training-free | DPS (Chung et al., 2022) | 1000 | 20.36 | 0.438 | 164.99 |
| | DDRM (Kawar et al., 2022) | 20 | 24.55 | 0.292 | 79.99 |
| | DDNM (Wang et al., 2022b) | 100 | 25.19 | 0.327 | 84.98 |
| | DiffPIR (Zhu et al., 2023) | 100 | 24.88 | 0.306 | 79.42 |
| | SITCOM (Alkhouri et al., 2024) | 20 | 24.79 | 0.277 | 61.88 |
| | CDDB (Chung et al., 2024) | 100 | 23.64 | 0.191 | 58.25 |
| Training-based | I2SB (Liu et al., 2023a) | 1000 | 23.36 | 0.178 | 60.99 |
| | I3SB (Wang et al., 2025) | 25 | 23.79 | 0.169 | 59.38 |
| | I3SB (Wang et al., 2025) | 1 | 25.24 | 0.157 | 124.47 |
| | DDC (Chen et al., 2024) | 5 | 24.67 | 0.156 | 62.06 |
| | ResShift (Yue et al., 2024b) | 4 | 23.68 | 0.207 | 60.75 |
| | SinSR (Wang et al., 2024c) | 1 | 22.25 | 0.207 | 94.90 |
| | Ours | 26 | 23.35 | **0.132** | **46.88** |
| | Ours distilled (t = 1) | 1 | 24.20 | 0.135 | 52.69 |
| | Ours distilled (t = 0.5) | 1 | 24.85 | 0.176 | 60.69 |
| | Ours distilled (t = 0) | 1 | **26.18** | 0.284 | 92.04 |

Table 4: Quantitative results of real-world image super-resolution on ImageNet 256×256 and RealSR (Cai et al., 2019). The best and second best results are highlighted in **bold** and underline. The number of inference steps is indicated by 's', which is the same as NFE when not use CFG.

| ImageNet | PSNR (↑) | LPIPS (↓) | FID (↓) | NIQE (↓) | MUSIQ (↑) | MANIQA (↑) | CLIPIQA (↑) |
|---|---|---|---|---|---|---|---|
| SwinIR (Liang et al., 2021) | 22.24 | 0.320 | 207.75 | 6.0084 | 47.46 | 0.5527 | 0.5544 |
| NAFNet (Chen et al., 2022) | 23.23 | 0.672 | 223.90 | 10.390 | 17.40 | 0.3025 | 0.3708 |
| ResShift-15s (Yue et al., 2024b) | 23.55 | 0.308 | 168.82 | 6.8026 | 49.95 | 0.5921 | 0.5906 |
| SinSR-1s (Wang et al., 2024c) | 23.19 | 0.302 | 157.10 | 6.1700 | 50.30 | 0.5789 | 0.5995 |
| Ours-15s | 22.51 | 0.308 | 153.34 | 5.9657 | **54.90** | **0.6151** | 0.6014 |
| Ours distilled-1s (t = 1) | 22.24 | 0.292 | 151.39 | 5.4043 | 54.16 | 0.5892 | 0.6066 |
| Ours distilled-1s (t = 0.5) | 23.85 | 0.396 | 201.77 | 8.9631 | 40.46 | 0.4905 | 0.4176 |
| Ours distilled-1s (t = 0) | 23.95 | 0.486 | 235.44 | 10.3695 | 35.04 | 0.3610 | 0.2886 |
| **RealSR** | **PSNR (↑)** | **LPIPS (↓)** | **FID (↓)** | **NIQE (↓)** | **MUSIQ (↑)** | **MANIQA (↑)** | **CLIPIQA (↑)** |
| ResShift-15s (Yue et al., 2024b) | 26.26 | 0.347 | 142.57 | 7.1780 | 58.47 | 0.5343 | 0.5481 |
| SinSR-1s (Wang et al., 2024c) | 26.27 | 0.321 | **137.59** | 6.2773 | 60.84 | 0.5418 | **0.6224** |
| Ours-15s | 25.41 | 0.297 | 145.34 | 4.9089 | **65.48** | **0.5705** | 0.5826 |
| Ours distilled-1s (t = 1) | 25.27 | **0.288** | 142.38 | 4.6337 | 65.30 | 0.5604 | 0.5891 |
| Ours distilled-1s (t = 0.5) | 26.76 | 0.311 | 175.11 | 6.9517 | 57.44 | 0.4879 | 0.4251 |
| Ours distilled-1s (t = 0) | **27.01** | 0.331 | 190.63 | 8.1201 | 53.09 | 0.4205 | 0.3129 |

| DAVI (1) | Ours (1) | I2SB (1000) | CDDB (100) | Ours (26) | DDC (5) | ResShift (4) | Ours (1) |

Figure 5: **Qualitative comparison with training-based methods.** The first two columns demonstrate 4× SR results on the FFHQ dataset with noise level $\sigma_n = 0.05$. The remaining columns show noiseless 4× SR results on the ImageNet dataset. Numbers next to the method names represent the required NFEs.

**Training Details.** We do experiments for both noisy and noiseless SR. For noiseless SR, bicubic downsampling is performed on all three datasets. For noisy SR, we conduct experiment only on FFHQ 256×256 dataset with average-pooling downsampling and Gaussian noise with a standard deviation $\sigma_y = 0.05$. All images are normalized to the range of $[-1, 1]$. For experiments on FFHQ 256×256 and DIV2K, we adopt the same model architecture used for FFHQ in (Chung et al., 2022); and for experiment on ImageNet 256×256, we use the same model architecture as the pre-trained unconditional model used in (Dhariwal & Nichol, 2021). We modify the input convolution layer to accept concatenated image input. The first stage models are trained from scratch and are sampled with RK45 sampler by default. The one-step model is initialized from the teacher model for distillation. We use the Adam optimizer with a linear warmup schedule over 1k training steps, followed by a learning rate of $1e-4$ for both stages.

## 4.2 RESULTS

**Quantitative Results.** We present comprehensive quantitative evaluations on several benchmark datasets: DIV2K, FFHQ, ImageNet and real world test set and different tasks (including noiseless SR, noisy SR and real world SR) (Tabs. 1 to 6). Our analysis reveals several findings: (i) The first-stage OFTSR achieves superior performance in perceptual metrics (FID and LPIPS) while requiring fewer than 32 NFEs. (ii) Our distillation

Table 5: Quantitative comparison on real world sets. The best and second best results are in **bold** and underline.

| Datasets | Method | NIQE ↓ | MUSIQ ↑ | MANIQA ↑ | CLIPIQA ↑ | LIQE ↑ |
|---|---|---|---|---|---|---|
| RealSet80 | SwinIR | **4.1601** | 63.72 | 0.5444 | 0.5919 | 3.6479 |
| | NAFNet | 8.8794 | 35.16 | 0.3975 | 0.5289 | 1.0969 |
| | ResShift-15s | 6.1955 | 61.35 | 0.5318 | 0.6702 | 3.4473 |
| | SinSR-1s | 5.6182 | 63.96 | 0.5376 | **0.7242** | 3.6072 |
| | Ours-15s | 4.3713 | 66.90 | **0.5617** | 0.6797 | 3.9982 |
| | Ours distilled-1s (t = 1) | 4.1826 | 67.46 | 0.5570 | 0.6904 | 4.0168 |
| RealLQ250 | SwinIR | 4.1628 | 60.48 | 0.5104 | 0.5352 | 3.0883 |
| | NAFNet | 9.5524 | 25.97 | 0.3360 | 0.4095 | 1.0512 |
| | ResShift-15s | 6.5731 | 59.98 | 0.5003 | 0.6239 | 2.9340 |
| | SinSR-1s | 5.8200 | 63.73 | 0.5161 | **0.6990** | 3.2578 |
| | Ours-15s | 4.2848 | 67.15 | **0.5481** | 0.6520 | 3.8367 |
| | Ours distilled-1s (t = 1) | 4.0731 | 67.32 | 0.5287 | 0.6532 | 3.7211 |

algorithm is versatile, when applied to ResShift (Yue et al., 2024b) teacher, our distilled model achieved better one-step performance than SinSR (Wang et al., 2024c) (see Tab. 9). (iii) Our distilled version of OFTSR demonstrates remarkable versatility, achieving either the highest PSNR

Table 6: Quantitative comparison of state-of-the-art one-step SR methods on synthetic (DIV2K-Val) and real-world (RealLQ250) benchmarks. Best results are in **bold**, second best are underlined. Our method is tested under $t = 1$. ResShift* means we train our noise-augmented conditional flow in Sec. 3.1 using the ResShift model architecture then distillation; and DiT4SR means we use the pre-trained DiT4SR model as the teacher model for distillation.

| Dataset | Metric | SinSR-1s | CTMSR-1s | AddSR-1s | OSEDiff-1s | TSDSR-1s | Ours (ResShift*)-1s | Ours (DiT4SR)-1s |
|---|---|---|---|---|---|---|---|---|
| **DIV2K-Val** | PSNR ↑ | 24.50 | **24.87** | 22.39 | 23.86 | 22.17 | 23.91 | 22.80 |
| | SSIM ↑ | 0.6136 | **0.6349** | 0.5652 | 0.6233 | 0.5680 | 0.6073 | 0.5774 |
| | LPIPS ↓ | 0.3164 | 0.3011 | 0.3728 | 0.2896 | **0.2679** | 0.3226 | 0.2716 |
| | DISTS ↓ | 0.2110 | 0.2102 | 0.2387 | 0.1999 | 0.1901 | 0.2081 | **0.1889** |
| | FID ↓ | 131.96 | 126.49 | 133.78 | 100.53 | 103.49 | 133.30 | **98.27** |
| | NIQE ↓ | 6.1721 | 5.3036 | 5.9929 | 4.9741 | **4.6621** | 4.9061 | 4.8399 |
| | MUSIQ ↑ | 64.26 | 66.59 | 63.39 | 68.53 | **71.19** | 68.71 | 70.25 |
| | MANIQA ↑ | 0.5442 | 0.5146 | 0.5657 | 0.6010 | 0.6010 | 0.5464 | **0.6145** |
| | CLIPIQA ↑ | 0.6687 | 0.6602 | 0.5734 | 0.6692 | 0.7221 | 0.6545 | **0.7233** |
| **RealLQ250** | NIQE ↓ | 5.8200 | 4.5835 | 4.9235 | 3.9656 | **3.4868** | 4.0731 | 3.7802 |
| | MUSIQ ↑ | 63.73 | 68.00 | 66.82 | 69.55 | 72.09 | 67.32 | **72.60** |
| | MANIQA ↑ | 0.5161 | 0.5078 | 0.5304 | 0.5782 | 0.5829 | 0.5287 | **0.5904** |
| | CLIPIQA ↑ | 0.6990 | 0.6706 | 0.6437 | 0.6725 | 0.7221 | 0.6532 | **0.7252** |
| | LIQE ↑ | 3.2578 | 3.3373 | 3.4929 | 3.9039 | 4.0834 | 3.7211 | **4.1122** |

scores or ranking among the top two methods for FID and LPIPS metrics in one step. This indicates minimal performance degradation between the teacher and student models. (iv) Our experiments suggest that FID serves as a more reliable indicator of perceptual quality and better captures the performance gap between teacher and student models during distillation. (v) When applied to a powerful SD-based SR model (DiT4SR), our distillation algorithm produces a one-step generator whose performance is competitive with other leading SOTA distillation methods. This also validates the versatility of our distillation algorithm.

**Visual Results.** Our experimental results demonstrate that OFTSR achieves high-quality image reconstructions. We evaluate OFTSR against leading training-free methods for $4\times$ SR, as shown in Fig. 4. While DPS can produce sharp reconstructions, it requires 1000 NFEs and often introduces significant distortions. In contrast, OFTSR successfully preserves structural information from low-resolution inputs while reconstructing fine details. Notably, our distilled version of OFTSR requires only one NFE, as other training-free methods suffer from severe error accumulation when using less than 10 NFEs. As illustrated in Fig. 5, we also compare OFTSR against state-of-the-art SR methods that require training. The results show that our approach generates patterns with rich, natural details. Furthermore, our distilled model enables flexible control over the fidelity-realism trade-offs in the generated high-resolution images. Fig. 3 demonstrates this capability through examples of noisy $4\times$ SR with varying degrees of realism and fidelity. More qualitative comparison and visual examples can be found in Sec. K

## 4.3 ABLATIONS

**Perturbation Strength $\sigma_p$.** In Tab. 7, we evaluate the design choices in the simple conditional flow training stage. All experiments in this ablation study are conducted under identical training conditions, with performance metrics measured using the RK45 solver. The most critical hyper-parameter in this ablation is the strength of the perturbation $\sigma_p$. Consistent with previous works, we confirm that perturbation is essential for generating perceptually compelling images from LR inputs. Notably, we discover that increasing perturbation strength does not necessarily improve perceptual quality but instead leads to more curved PF-ODE, requiring additional NFEs to solve (see Tab. 7). Furthermore, our experiments demonstrate that conditioning on $\mathbf{x}_{LR}$ is crucial to compensate for information loss during perturbation. We also find that $\ell_1$ loss outperforms $\ell_2$ loss for our specific task. While (Kim et al., 2024) previously highlighted the significance of Gaussian perturbation, our work is the first to systematically analyze the relationship between noise perturbation and the trade-off between generation quality and efficiency in flow-based models.

**Distillation Design Space.** In Tab. 8, we evaluate several crucial design choices for the distillation stage, including the distillation loss type, solver type, d$t$ value, and the weighting of alignment and boundary losses. Since learning $\mathbf{v}_\phi(\mathbf{x}_{0,LR}, 0)$ is considerably easier than learning $\mathbf{v}_\phi(\mathbf{x}_{0,LR}, 1)$, we utilize metrics from the latter to decide our distillation hyperparameters. Our analysis of the step size

Table 7: Ablation on noiseless FFHQ 256×256 first stage. The default training setting is bs = 32; lr = 0.0001; loss type = $\ell_1$; with condition; all experiments are trained for 100k steps. The final choice is highlighted to balance the performance and efficiency.

| Strength of Perturbation $\sigma_p$ | NFEs (↓) | PSNR (↑) | LPIPS (↓) | FID (↓) |
|---|---|---|---|---|
| 0. | 20 | 29.04 | 0.244 | 110.29 |
| 0.001 | 20 | 29.56 | 0.115 | 48.39 |
| 0.01 | 20 | 29.56 | 0.066 | 34.70 |
| 0.1 | 20 | 28.83 | 0.053 | 30.54 |
| 0.2 | 27 | 28.84 | 0.053 | 30.77 |
| 0.3 | 32 | 28.88 | 0.053 | 31.02 |
| 0.5 | 32 | 28.86 | 0.053 | 30.22 |
| 0.8 | 44 | 28.84 | 0.054 | 31.02 |
| 1. | 44 | 28.82 | 0.053 | 30.75 |
| 0.1 (no cond) | 20 | 28.09 | 0.073 | 42.47 |
| 0.1 ($\ell_2$) | 20 | 28.60 | 0.055 | 31.86 |

Table 8: Ablation on noiseless FFHQ 256×256 distillation stage. The default training setting is bs = 8; $\sigma_p = 0.1$, lr = 0.0001; loss type = $\ell_2$; with LR condition; all experiments are trained for 20k steps; And the one-step metrics are calculated with $t = 1$. Ablations in subgroups can be ordered as $dt \rightarrow \lambda_{\text{BC}} \rightarrow \lambda_{\text{align}} \rightarrow$ Solver, and $dt \rightarrow$ Distillation Loss.

| Distillation Loss | Solver | $dt$ | $\lambda_{\text{align}}$ | $\lambda_{\text{BC}}$ | PSNR (↑) | LPIPS (↓) |
|---|---|---|---|---|---|---|
| Ours | Euler | 0.001 | 0 | 0 | 28.77 | 0.160 |
| Ours | Euler | 0.01 | 0 | 0 | 29.35 | 0.076 |
| Ours | Euler | 0.02 | 0 | 0 | 29.48 | 0.068 |
| Ours | Euler | 0.05 | 0 | 0 | 29.73 | 0.065 |
| Ours | Euler | 0.1 | 0 | 0 | 30.05 | 0.073 |
| BOOT (Gu et al., 2023) | Euler | 0.05 | 0 | 0 | 23.81 | 0.483 |
| PINN (Tee et al., 2024) | Euler | 0.05 | 0 | 0 | 27.92 | 0.250 |
| Ours | Euler | 0.05 | 0 | 0.1 | 29.73 | 0.064 |
| Ours | Euler | 0.05 | 0.01 | 0.1 | 29.69 | 0.063 |
| Ours | Heun | 0.05 | 0.01 | 0.1 | 29.21 | 0.057 |
| Ours | Ralston | 0.05 | 0.01 | 0.1 | 29.15 | 0.056 |
| Ours | Midpoint | 0.05 | 0.01 | 0.1 | 29.14 | 0.056 |
| Ours (bs=32) | Midpoint | 0.05 | 0.01 | 0.1 | 29.07 | 0.055 |

$dt$ reveals that smaller values do not necessarily yield better results, leading us to select $dt = 0.05$ for subsequent experiments. Our proposed loss function demonstrates substantial improvement over both the original BOOT (Gu et al., 2023) loss and PINN (Tee et al., 2024) distillation loss, achieving a significant LPIPS score improvement of more than 0.1. Further experimentation shows that both the alignment loss (Eq. (10)) and boundary loss (Eq. (11)) contribute to enhanced performance. By combining these losses with a Midpoint 2-order solver, we achieve additional improvements in our one-step model's performance at $t = 1$.

## 4.4 COMPUTATIONAL OVERHEAD

**Training Cost Comparison.** Our distillation algorithm is highly flexible and can be easily applied to any pre-trained diffusion/flow-based conditional model. As shown in Tab. 9, we applied our distillation algorithm to the ResShift(Yue et al., 2024b) pre-trained model and achieved teacher-level performance in one step, surpassing SinSR(Wang et al., 2024c) in FID with much less training compute. Even taking the training stage into account with a larger model, our method remains more efficient than ResShift. We use $t = 1$ for OFTSR.

Table 9: Comparison of training cost on single NVIDIA A100.

| Method | [NFE(↓)] | # Iterations | s/Iter | Training Time | PSNR (↑) | LPIPS (↓) | FID (↓) |
|---|---|---|---|---|---|---|---|
| DDC (Chen et al., 2024) (base model frozen) [5] | | 160k | 0.89 | ~1.65 days | **24.67** | 0.156 | 62.06 |
| ResShift (Yue et al., 2024b) (teacher) [4] | | 500k | 1.32 | ~7.64 days | 23.68 | 0.207 | 60.75 |
| SinSR (Wang et al., 2024c) (ResShift teacher) [1] | | 30k | 7.41 | ~2.57 days | 22.25 | 0.207 | 94.90 |
| OFTSR (ResShift teacher) [1] | | 5k | 6.72 | **~0.39 days** | 24.01 | 0.218 | 60.64 |
| OFTSR (pre-train+distill) [1] | | 100k + 50k | 3.9/4.4 | ~4.51+2.54 days | 24.20 | **0.135** | **52.69** |

**Inference Cost Comparison.** We have included a detailed comparison of the inference cost in Tab. 10, using FLOPS and MAC to measure model complexity. We use $t = 1$ for OFTSR.

Table 10: Comparison of inference cost on single NVIDIA A100.

| Method | [NFE(↓)] | # Params (+VAE) | FID (↓) | MACs (+VAE) (↓) | FLOPs (+VAE) (↓) | Runtime (↓) |
|---|---|---|---|---|---|---|
| DDNM (Wang et al., 2022b) [100] | | 552.8M | 84.98 | 1.11T | 2.24T | 7.00s |
| DDC (Chen et al., 2024) [5] | | 552.8+113.7M | 62.06 | 1.11+0.24T | 2.24+0.49T | 0.74s |
| ResShift (Yue et al., 2024b) [4] | | 118.6+55.3M | 60.75 | 50.1+473.5G | 100.4+948.5G | 0.27s |
| SinSR (Wang et al., 2024c) [1] | | 118.6+55.3M | 94.90 | 50.1+473.5G | 100.4+948.5G | **0.09s** |
| OFTSR (ResShift teacher) [1] | | 118.6+55.3M | 60.64 | 50.1+473.5G | 100.4+948.5G | **0.09s** |
| OFTSR (pre-train+distill) [1] | | 552.8M | **52.69** | 1.11T | 2.24T | 0.21s |

## 5 CONCLUSION

In this paper, we introduced OFTSR, a novel approach to developing efficient one-step image super-resolution models. Our extensive experiments on FFHQ, DIV2K, ImageNet and real world SR datasets demonstrate that our method significantly improves computational efficiency while maintaining high-quality image restoration capabilities. The proposed framework represents a promising direction in efficient image SR, effectively addressing the perception-distortion trade-off.

ACKNOWLEDGEMENTS

**Acknowledgments:** This work was supported by the National Natural Science Foundation of China (Grant No. 62572234), the Gusu Innovation and Entrepreneurship Leading Talent Program (Grant No. ZXL20254324), and the Suzhou Key Technologies Project (Grant No. SGY2023136).

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

# Appendix for OFTSR

## A  USE OF LARGE LANGUAGE MODELS

We used large language models solely for text polishing and grammar correction during manuscript preparation. No LLMs were involved in the conception or design of the method, experiments, or analysis. All technical content, results, and conclusions have been independently verified and validated by the authors.

## B  RELEVANT DERIVATIONS TO OUR DISTILLATION LOSS

### B.1  CONTINUOUS VERSION OF THE FINAL LOSS

We provide detailed derivation to our distillation loss used in the paper. By substitute intermediate results $\mathbf{x}_s$ and $\mathbf{x}_t$ from student model Eq. (7) into the ODE induced by teacher model Eq. (8), we have:

$$
\begin{aligned}
\cancel{\mathbf{x}_0} + s\mathbf{v}_\phi(\mathbf{x}_{0,\text{LR}}, s) &= \cancel{\mathbf{x}_0} + t\mathbf{v}_\phi(\mathbf{x}_{0,\text{LR}}, t) + (s-t)\mathbf{v}_\theta(\mathbf{x}_{t,\text{LR}}, t) \\
\Longrightarrow s(\mathbf{v}_\phi(\mathbf{x}_{0,\text{LR}}, s) - \mathbf{v}_\phi(\mathbf{x}_{0,\text{LR}}, t)) &= (t-s)\mathbf{v}_\phi(\mathbf{x}_{0,\text{LR}}, t) + (s-t)\mathbf{v}_\theta(\mathbf{x}_{t,\text{LR}}, t) \\
&= \mathrm{d}t(\mathbf{v}_\theta(\mathbf{x}_{t,\text{LR}}, t) - \mathbf{v}_\phi(\mathbf{x}_{0,\text{LR}}, t)).
\end{aligned}
\tag{14}
$$

Start from this constraint that applies to the student model, we can construct distillation loss in different forms. (i) In the same spirit as BOOT (Gu et al., 2023), we make only $\mathbf{v}_\phi(\mathbf{x}_{0,\text{LR}}, s)$ and this will lead to loss Eq. (12). (ii) If we only detach the teacher output, we will end up with loss similar to PINN based distillation PID proposed in (Tee et al., 2024):

$$
\mathcal{L}_{\text{PINN}}(\phi) := \mathbb{E}_{\mathbf{x}_1 \sim p_1, t \sim \mathcal{U}[0,1]} \left[ \left\| \left[ \frac{s}{\mathrm{d}t} \left( \mathbf{v}_\phi(\mathbf{x}_{0,\text{LR}}, s) - \mathbf{v}_\phi(\mathbf{x}_{0,\text{LR}}, t) \right) + \mathbf{v}_\phi(\mathbf{x}_{0,\text{LR}}, t) \right] - \text{SG}\left[ \mathbf{v}_\theta(\mathbf{x}_{t,\text{LR}}, t) \right] \right\|_2^2 \right].
\tag{15}
$$

Both Eqs. (12) and (15) are loss variants from Eq. (8), and we did not try other variant given the already-good performance of Eq. (12).

In addition, by considering the intermediate interpolation $\mathbf{x}_t = (1-t)\mathbf{x}_0 + t\mathbf{x}_1$ as a special case of $\mathbf{x}_t = \sigma_t \mathbf{x}_0 + \alpha_t \mathbf{x}_1$ in BOOT (Gu et al., 2023), we can derive the following distillation loss:

$$
\mathcal{L}_{\text{BOOT}}(\phi) := \mathbb{E}_{\mathbf{x}_1 \sim p_1, t \sim \mathcal{U}[0,1]} \left[ \frac{1}{\lambda^2} \left\| \mathbf{x}_\phi(\mathbf{x}_{0,\text{LR}}, s) - \text{SG}\left[ \mathbf{x}_\phi(\mathbf{x}_{0,\text{LR}}, t) + \lambda\left(\mathbf{x}_\theta(\mathbf{x}_{t,\text{LR}}, t) - \mathbf{x}_\phi(\mathbf{x}_{0,\text{LR}}, t)\right) \right] \right\|_2^2 \right],
\tag{16}
$$

where $\lambda = 1 - \frac{t(1-s)}{s(1-t)}$, $\mathbf{x}_\phi(\mathbf{x}_{0,\text{LR}}, t) = \mathbf{x}_0 + \mathbf{v}_\phi(\mathbf{x}_{0,\text{LR}}, t)$, $\mathbf{x}_\phi(\mathbf{x}_{0,\text{LR}}, s) = \mathbf{x}_0 + \mathbf{v}_\phi(\mathbf{x}_{0,\text{LR}}, s)$, and $\mathbf{x}_\theta(\mathbf{x}_{t,\text{LR}}, t) = \mathbf{x}_t + (1-t)\mathbf{v}_\theta(\mathbf{x}_{t,\text{LR}}, t)$ with $\mathbf{x}_t = \mathbf{x}_0 + t\mathbf{v}_\phi(\mathbf{x}_{0,\text{LR}}, t)$. We compared our proposed loss Eq. (12) with its variant Eq. (15) and Eq. (16) in Tab. 8 and our ablation shows that Eq. (12) works best for SR task.

### B.2  OFTSR AS FORWARD DISTILLATION

The general form of our OFTSR loss or BOOT (Gu et al., 2023) loss can be seen as a special case of forward distillation (Boffi et al., 2025; Sabour et al., 2025; Liu, 2025). Start from general relation:

$$
\mathbf{x}_t + (s-t)\mathbf{v}_\phi(\mathbf{x}_t, t, s) = \mathbf{x}_s.
\tag{17}
$$

where $\mathbf{v}_\phi(\mathbf{x}_t, t, s)$ is the mean velocity defined on the time interval $[t, s]$ as defined in MeanFlow (Geng et al., 2025).

The MeanFlow loss can be derived directly by taking derivative w.r.t. $t$ of Eq. (17), which is also named as backward distillation loss. Similarly, when taking derivative w.r.t. $s$, the end timestep of the interval, of Eq. (17), we get the forward distillation loss.

For our OFTSR loss, if we consider mapping from arbitrary start timestep $t$ to two close end timestep $s_1$ and $s_2$, and connecting the corresponding two state $\mathbf{x}_{s_1}$ and $\mathbf{x}_{s_2}$, we have:

$$
\begin{aligned}
\cancel{\mathbf{x}_t} + (s_2 - t)\mathbf{v}_\phi(\mathbf{x}_t, t, s_2) &= \cancel{\mathbf{x}_t} + (s_1 - t)\mathbf{v}_\phi(\mathbf{x}_t, t, s_1) + (s_2 - s_1)\mathbf{v}_\theta(\mathbf{x}_{s_1}, s_1) \\
\Longrightarrow (s_2 - t)(\mathbf{v}_\phi(\mathbf{x}_t, t, s_2) - \mathbf{v}_\phi(\mathbf{x}_t, t, s_1)) &= (s_2 - s_1)(\mathbf{v}_\theta(\mathbf{x}_{s_1}, s_1) - \mathbf{v}_\phi(\mathbf{x}_t, t, s_1)).
\end{aligned}
\tag{18}
$$

For $s_2 - s_1 = \mathrm{d}s$ and $\lim_{\mathrm{d}s \to 0}$, we have $s_1 = s_2 = s$ and:

$$(s - t)\frac{\mathrm{d}}{\mathrm{d}s}\mathbf{v}_\phi(\mathbf{x}_t, t, s) = \mathbf{v}_\theta(\mathbf{x}_s, s) - \mathbf{v}_\phi(\mathbf{x}_t, t, s), \tag{19}$$

which recovers the forward distillation loss as the time derivative w.r.t. $s$ of Eq. (17). Thus we can view the OFTSR loss and BOOT (Gu et al., 2023) loss as a discretization of the forward distillation loss. And it is easy to verify that the signal ODE in BOOT is equivalent to our distillation loss in the flow schedule.

## C  LIMITATIONS

While our method advances one-step image super-resolution, limitations include performance constrained by teacher model capabilities. Future work will incorporate ground-truth supervision through regression loss or adversarial training.

## D  DIFFUSION AND PERCEPTION-DISTORTION TRADE-OFF

In practice, we found that our distilled model is slightly off the perception-distortion frontier of the teacher model, as displayed in Fig. 7. To be specific, the corresponding timestep $t$ shifts a bit but for the same MMSE value the first-stage model and distilled model have very close LPIPS value. This might be caused by the error from large step size $\mathrm{d}t$ used in practice and we leave this for future investigation.

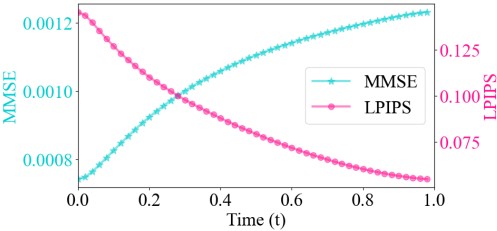

Figure 6: Metrics evaluation of estimated $\mathbf{x}_1^t$ across different timesteps $t$. During sampling, at each timestep $t$, we estimate the final image $\mathbf{x}_1^t$ using the current model prediction $\mathbf{v}_\theta(\mathbf{x}_{t,\mathrm{LR}}, t)$ and state $\mathbf{x}_t$ via $\mathbf{x}_1^t = \mathbf{x}_t + (1 - t)\mathbf{v}_\theta(\mathbf{x}_{t,\mathrm{LR}}, t)$. Both MMSE and LPIPS metrics are averaged over 100 sampling processes. We present MMSE instead of PSNR for better visual effect.

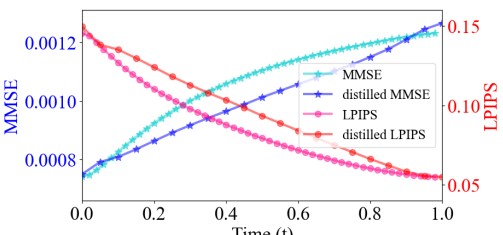

Figure 7: Metrics evaluation of estimated $\mathbf{x}_1^t$ across different timesteps $t$ for both teacher model and distilled one-step model. The teacher model is the same as the one in Fig. 6. We present MMSE instead of PSNR for better visual effect.

## E  MORE EXPERIMENTAL DETAILS

The training of all networks across both stages is smoothed using Exponential Moving Average (EMA) with a ratio of 0.9999. For FFHQ and ImageNet datasets, images are resized to 256 pixels with center cropping, while DIV2K training employs random crops of 256×256 patches. Data augmentation consists of horizontal flips with 50% probability and vertical flips with 6% probability throughout all experiments. For FFHQ noiseless experiment, we use default perturbation std $\sigma_p = 0.1$; for FFHQ noisy experiment, we use a higher perturbation std $\sigma_p = 0.5$ to cover the resized noise from LR images, as suggested in Tab. 11; for both DIV2K and ImageNet we use $\sigma_p = 0.2$. For training, we employed three widely-used datasets: the standard ImageNet training set (1.28M images), the DIV2K training set (800 2K resolution images), and a subset of FFHQ consisting of the first 60,000 images from the dataset. All models are trained until convergence or up to 300k training iterations and we select the model based on best metrics. We train the model with uniform loss weight on $t$. In the distillation stage, we sample the timestep $t$ using $t \sim \mathcal{U}\left[t_{\min}, t_{\max}\right]$ with $t_{\min} = 0.01$ and $t_{\max} = 0.99$ in practice.

For DIV2K evaluation, we first segment the large 2K resolution images into 256×256 patches for model inference, then reconstruct the final image by combining the restored patches. To ensure fair

---

**Algorithm 1** OFTSR Distillation

---

**Require:** teacher flow $\mathbf{v}_\theta$, dataset $\mathcal{D}_{\text{HR}}$, $\sigma_n$, $\sigma_p$, $\mathrm{d}t$, $w(t)$
 1: Initialize the one-step student $\mathbf{v}_\phi$ with the weights of $\mathbf{v}_\theta$
 2: **repeat**
 3:     Randomly sample $\mathbf{x}_1 \sim \mathcal{D}_{\text{HR}}$
 4:     Randomly sample $\mathbf{n} \sim \mathcal{N}(\mathbf{0}, \sigma_n \mathbf{I})$; $\mathbf{n}_p \sim \mathcal{N}(\mathbf{0}, \sigma_p \mathbf{I})$
 5:     Compute $\mathbf{x}_{\text{LR}} = \mathcal{H}^T(\mathcal{H}(\mathbf{x}_1) + \mathbf{n})$   // LR condition
 6:     Compute $\mathbf{x}_0 = \sqrt{1 - \sigma_p^2}\mathbf{x}_{\text{LR}} + \sigma_p \mathbf{n}_p$
 7:     Sample $t \in \mathcal{U}[0, 1]$ and $s = t + \mathrm{d}t$
 8:     Generate velocities $\mathbf{v}_\phi(\mathbf{x}_{0,\text{LR}}, t)$ and $\mathbf{v}_\phi(\mathbf{x}_{0,\text{LR}}, s)$
 9:     Calculate $\mathbf{x}_{t,\text{LR}} = \mathbf{x}_0 + t\mathbf{v}_\phi(\mathbf{x}_{0,\text{LR}}, t)$ and generate velocity $\mathbf{v}_\theta(\mathbf{x}_{t,\text{LR}}, t)$ by teacher model
10:     Compute $\mathcal{L}_{\text{distill}}$ with Eq. (9) and $\mathcal{L}_{\text{align}}$ with Eq. (10)
11:     Generate velocities $\mathbf{v}_\phi(\mathbf{x}_{0,\text{LR}}, 0)$ and $\mathbf{v}_\theta(\mathbf{x}_{0,\text{LR}}, 0)$ and compute $\mathcal{L}_{\text{BC}}$ with Eq. (11)
12:     Compute $\mathcal{L}(\phi) = \mathcal{L}_{\text{distill}}(\phi) + \lambda_{\text{align}}\mathcal{L}_{\text{align}}(\phi) + \lambda_{\text{BC}}\mathcal{L}_{\text{BC}}(\phi)$
13:     Optimize $\phi$ with a gradient-based optimizer using $\nabla_\phi \mathcal{L}$
14: **until** $\mathcal{L}(\phi)$ converges
15: **Return** one-step flow $\mathbf{v}_\phi$

---

Table 11: Ablation on FFHQ 256×256 first stage with noisy SR; default: bs = 32; $lr = 0.0001$; $\ell_1$ loss; with LR condition.

| $\sigma_p$ | NFEs ($\downarrow$) | PSNR ($\uparrow$) | LPIPS ($\downarrow$) |
|---|---|---|---|
| 0. | 20 | 25.23 | 0.319 |
| 0.1 | 20 | 24.09 | 0.158 |
| 0.3 | 32 | 24.14 | 0.154 |
| 0.5 | 32 | 24.10 | 0.154 |
| 1. | 44 | 24.22 | 0.153 |

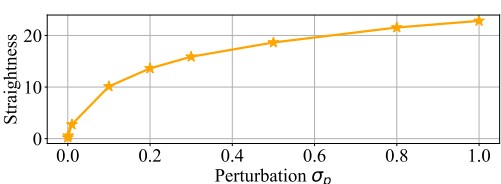

Figure 8: Straightness of conditional flows with different perturbation strength $\sigma_p$.

comparison, all generated SR images are stored in a dedicated separated folder with consistent file names across all evaluated methods, followed by metric calculation against the HR folder using our evaluation script. LPIPS scores are computed using the 'alex' model architecture. All experiments are conducted using 4 NVIDIA H800 GPUs.

The straightness of the learned flow $\mathbf{v}$ can be calculated with:

$$S(\mathbf{v}) = \int_0^1 \mathbb{E}\left[\|\mathbf{v}(\mathbf{x}_t, t) - (\mathbf{x}_1 - \mathbf{x}_0)\|^2\right]\mathrm{d}t, \tag{20}$$

We also measured the FID among 50k imagenet validation set and the result FID is 2.458 comparing to 2.8 from I2SB.

## F    ADDITIONAL EXPERIMENTS

We evaluated our first-stage training on the FFHQ 256×256 dataset using $\sigma_p = 1$ without conditioning, effectively training an unconditional generative model for human faces. For this experiment, we do not use any data augmentation. Our evaluation consists of generating 1k images from random noise using the RK45 sampler (with a ODE tolerance of 1e-3) and comparing them against the full training dataset of 70k images (we train our unconditional generative flow with the whole dataset). Initial experiments with $\ell_1$ loss yielded a FID score of 41.042 with an average of 56 NFEs, which falls short of the previous state-of-the-art P2 model's score of 28.139 (Choi et al., 2022). However, switching to $\ell_2$ loss for standard rectified flow training significantly improved performance, achieving a FID of 24.577 with only 44 NFEs on average. The model architecture used in our experiment is the same as the one used in P2. We leave further investigation to this discrepancy between $\ell_1$ and $\ell_2$ for image generation and restoration as future works. To facilitate a direct comparison with P2's best reported results (FID scores of 6.92 and 6.97 with 1,000 and 500 NFEs respectively (Choi et al., 2022)), we generated 50k samples using our $\ell_2$ loss-trained model. Our approach achieved a superior FID score of 5.871 with substantially fewer NFEs (44), demonstrating the effectiveness of rectified flow. Representative non-cherry-picking samples from our model are presented in Fig. 12.

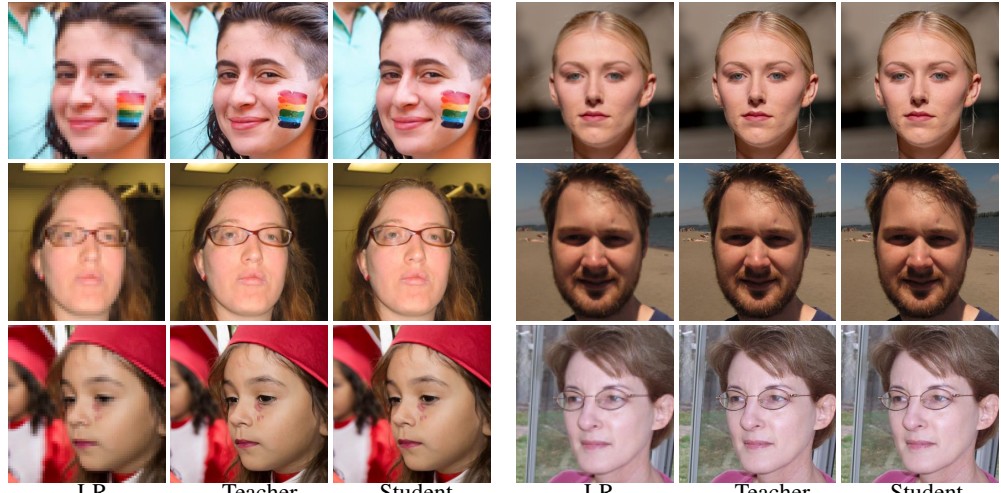

| LR | Teacher | Student | LR | Teacher | Student |

Figure 9: Visual results for $8\times$ (left) and $4\times$ (right) SR from resolution 64 to 512 and 128 to 512 respectively.

As our distillation technique is designed for image restoration tasks, we skip the distillation of this unconditional generation flow.

# G  ADDITIONAL RESULTS

## G.1  STRAIGHTNESS VS PERTURBATION STRENGTH

In Fig. 8, we validate the observation in Sec. 4.3 by also measuring the straightness of conditional flows. We observe that for SR task, the straightness is related to the noise perturbation added to the initial distribution, and a straighter flow does not lead to better performance.

## G.2  TRAINING DATASETS

In both stages of our approach, we utilize the same dataset. The following table shows comparable performance across different datasets for distillation with FFHQ teacher.

Table 12: Comparison of distilling FFHQ OFTSR teacher on FFHQ and Celeba-HQ dataset.

| | Distillation Dataset | Hyper-parameter $t$ | PSNR ($\uparrow$) | LPIPS ($\downarrow$) | FID ($\downarrow$) |
|---|---|---|---|---|---|
| FFHQ OFTSR Teacher | FFHQ | 1 | 28.98 | **0.055** | **36.02** |
| | Celeba-HQ | 1 | **29.75** | 0.056 | 41.25 |

## G.3  DIFFERENT RESOLUTION AND SCALE FACTOR (SF)

In this work, by default we follow previous works to use the setup of $4\times$ SR at $256 \times 256$. We also test $\text{SF} = 8$ on $256 \times 256$ and $\text{SF} = 4\&8$ on 512-resolution FFHQ, the results are shown in Tab. 13. All models are trained for 30k iterations (bs = 32) and distilled for 10k iterations (bs = 16). We visualize $8\times$ and $4\times$ reconstruction of teacher and student in Fig. 9.

Table 13: A comparison of the models trained across different resolution and scale factor.

| Method | Target Resolution | Scale Factor | NFE ($\downarrow$) | PSNR ($\uparrow$) | LPIPS ($\downarrow$) | FID ($\downarrow$) |
|---|---|---|---|---|---|---|
| DDNM (Wang et al., 2022b) | 256 | 8 | 100 | 25.65 | 0.178 | 104.47 |
| OFTSR (distilled) | 256 | 8 | 44 (1) | 25.74 (25.89) | 0.121(0.126) | 72.83 (93.74) |
| Unofficial SR3 (Saharia et al., 2022) | 512 | 8 | 2000 | 21.93 | 0.386 | 67.31 |
| OFTSR (distilled) | 512 | 8 | 32 (1) | 27.31 (28.12) | 0.151 (0.153) | 42.20 (42.33) |
| OFTSR (distilled) | 512 | 4 | 32 (1) | 30.80 (31.30) | 0.073 (0.072) | 13.21 (13.95) |

# H  FAILURE CASE

We show visualization of extreme t (boundary t and out of distribution t) in Fig. 10. Results of t ranges from [0,1] do not show failure case, while (ill-defined) OOD t, especially $t < 0$ fails.

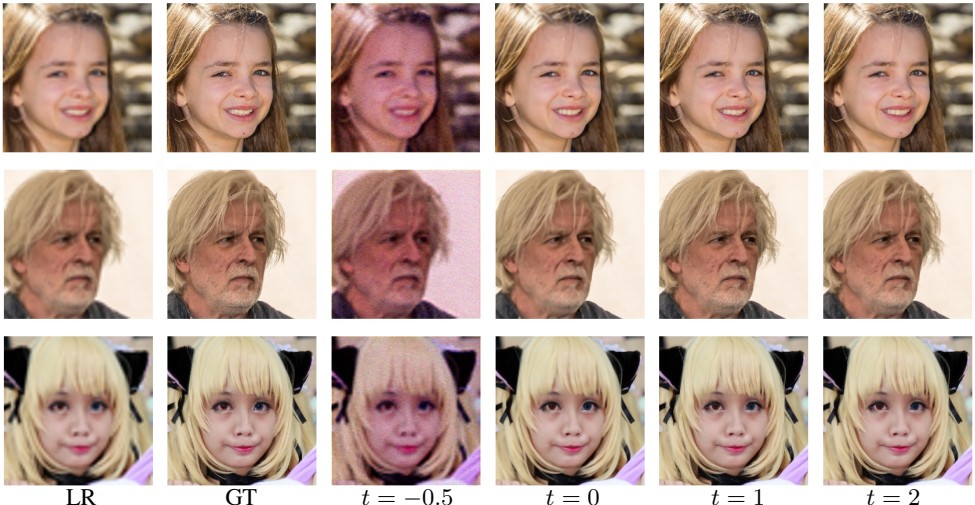

LR      GT      $t = -0.5$      $t = 0$      $t = 1$      $t = 2$

Figure 10: Visual results of boundary $t$ (0 and 1) and out of distribution $t$ (-0.5 and 2)

## I  RECONSTRUCTION DIVERSITY

The noise-augmented initialization (Sec. 3.1) introduces stochasticity that enables multiple diverse HR reconstructions for the same LR input. Both the teacher and student model can give different restorations of a LR image under different random seeds, and the visualization is shown in Fig. 11.

## J  THE CHOICE OF $t$

The parameter $t$ controls the fidelity–realism trade-off and is inherently guided by user preference: $t \approx 0$ favors maximal fidelity, while $t \approx 1$ emphasizes realism. In our experiments, the fidelity–realism parameter $t$ is not highly sensitive to the dataset or degradation type: its effective range stays consistent. When a target domain or evaluation metric is specified, $t$ can also be chosen automatically by optimizing it on a small validation set to best satisfy the desired objective or target. This enables both user-driven and metric-driven control of the fidelity–realism balance.

## K  ADDITIONAL VISUAL SAMPLES AND COMPARISONS

In this section, we present additional visual results that demonstrate our method's capabilities. Fig. 13 showcases multiple examples illustrating the tunable fidelity-realism trade-offs achieved on the FFHQ dataset. Figs. 14 and 15 provide comprehensive comparisons between our method and existing approaches on FFHQ and ImageNet images, respectively. In Fig. 16, we compare real-world (without synthetic degradation) SR results, under the $128 \rightarrow 512$ SR setting. In Fig. 17, we shows OFTSR can perform arbitrary scale SR. Here, the model is trained solely on ImageNet for $64 \rightarrow 256$ SR, demonstrating strong resolution and scale generalization without any retraining. Additionally, in Fig. 18, we demonstrate our method's performance on both real-world SR tasks and AI-generated content enhancement. In Figs. 19 and 20, we compare visually our OFTSR (DiT4SR) with other SOTA method for one-step large resolution SR. Results from Figs. 14, 15 and 18 are generated with our distilled one-step model unless otherwise specified.

## L  DISCUSSION OF ACCELERATED I2SB METHODS

We provide here a detailed discussion of recent accelerated variants of I2SB and clarify their relationship to our approach.

**I3SB (Wang et al., 2025).**   I3SB introduces an improved sampling algorithm for pretrained I2SB models, analogous to DDIM for DDPM. While it yields faster sampling and moderately better re-

sults, it retains the fundamental behavior of the original I2SB: multi-step sampling is required to achieve high perceptual realism, whereas a single step primarily preserves fidelity. In contrast, our distillation framework produces a one-step model that attains *much stronger realism* while also enabling a controllable fidelity–realism trade-off.

**CDBM (He et al., 2024).** CDBM proposes consistency bridge training and consistency bridge distillation for diffusion bridge models, mirroring the consistency-training paradigm used in consistency models. However, its experimental scope is limited to relatively small image-to-image translation tasks (e.g., Edges $\rightarrow$ Handbags, DIODE-Outdoor) and ImageNet inpainting. Since no SR evaluation or open-source implementation is provided, direct comparison in our setting is not feasible.

**IBMD (Gushchin et al., 2025).** IBMD introduces a distributional matching algorithm for conditional bridge models, conceptually related to DMD (Yin et al., 2024b) for continuous diffusion models. The method requires learning an additional auxiliary network, which increases computational overhead and can introduce training instability. Moreover, reported results show that the one-step performance of IBMD is comparable to I2SB with 1000 NFEs, suggesting limited advantages for efficient one-step SR. Therefore, our distilled model remains competitive or superior in the one-step regime.

Overall, while these works explore acceleration or distillation within the I2SB family, they differ substantially in objectives, model scope, and applicability to super-resolution. Our approach provides a reproducible and effective one-step SR framework with controllable fidelity–realism behavior not addressed in prior I2SB variants.

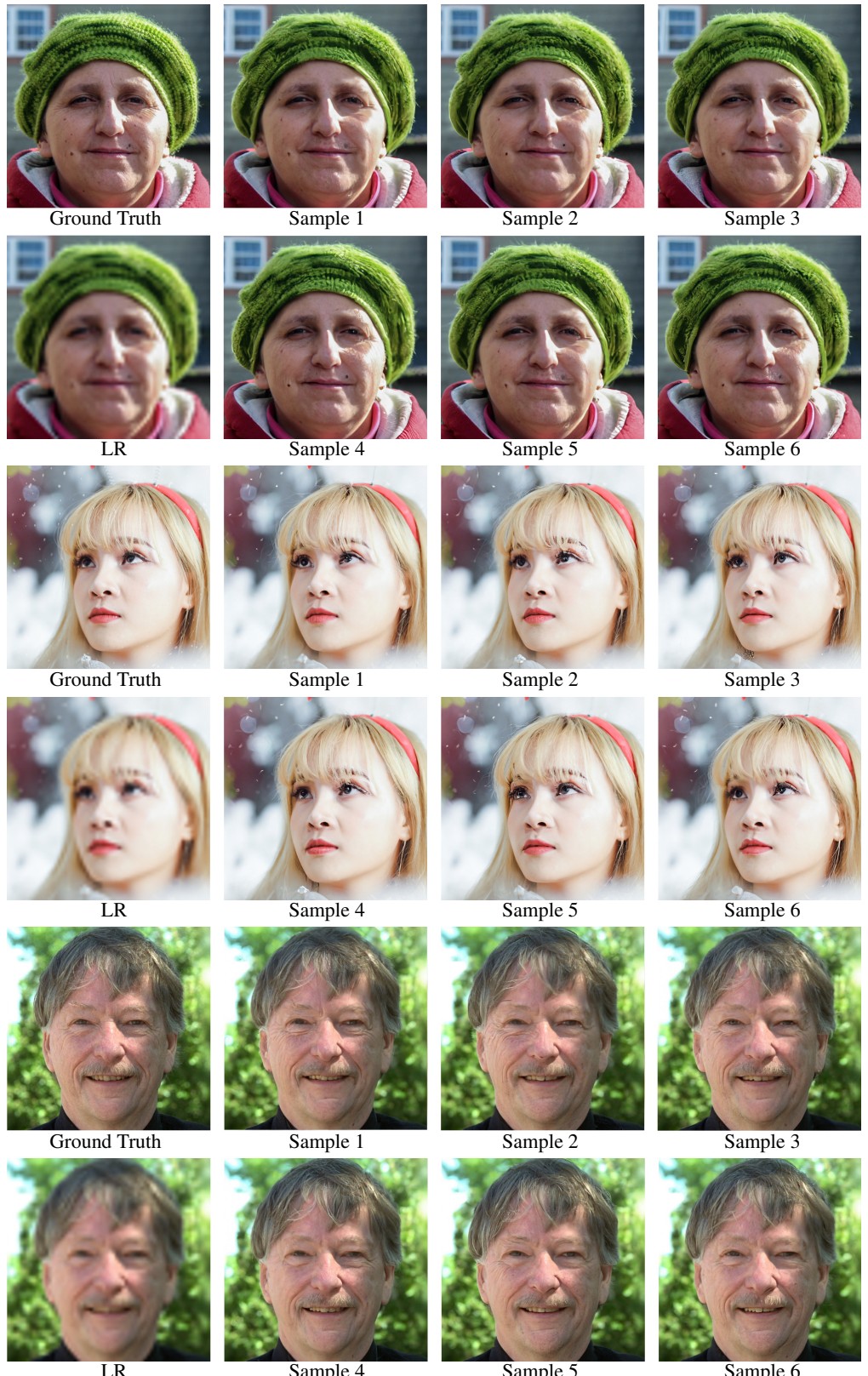

Figure 11: Our method can maintain diversity in outputs. The resolution of ground truth image is $512 \times 512$ and the LR is $64 \times 64$. The first group is generated by the teacher model, while the remaining two groups are produced by the student model.

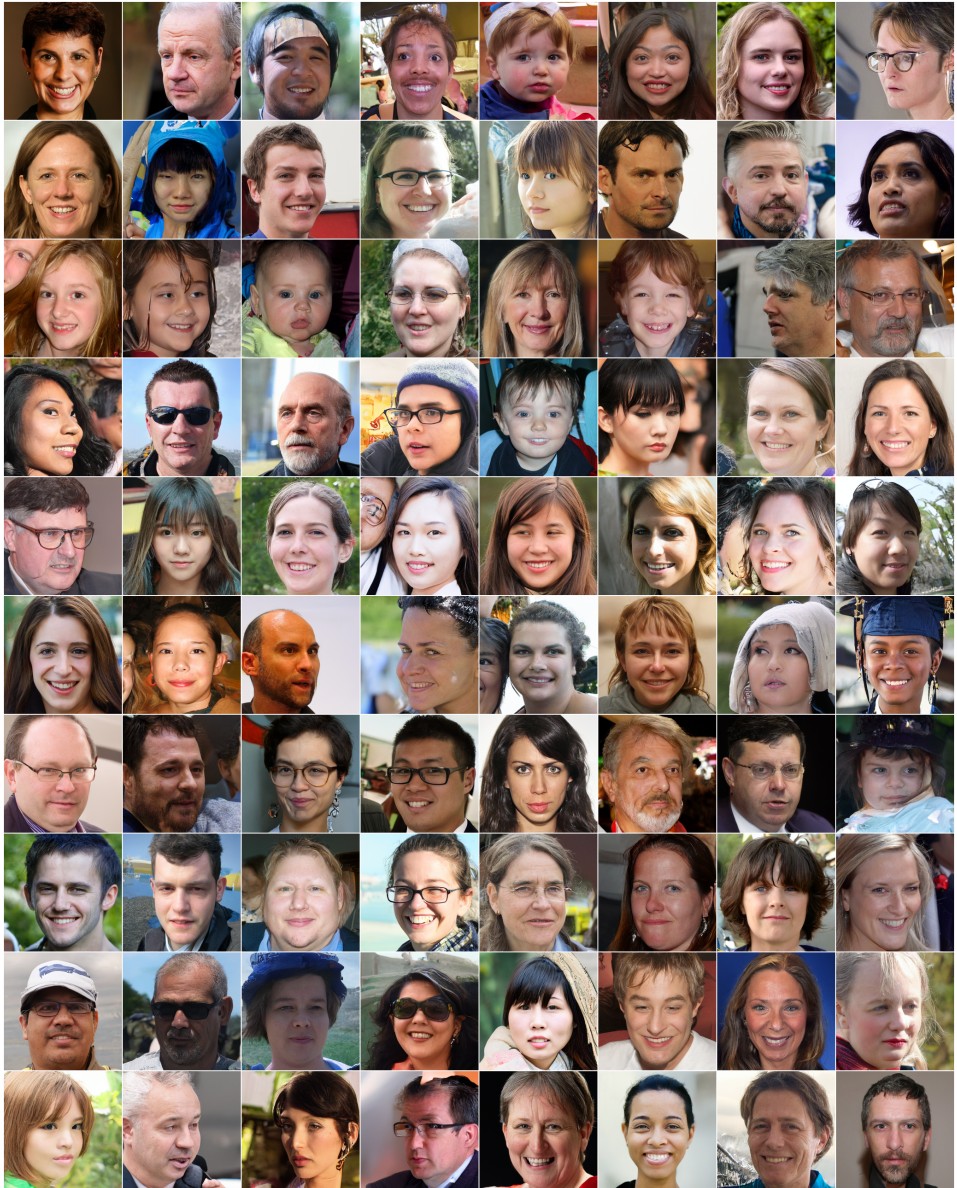

Figure 12: Random generated samples from unconditional model trained on FFHQ dataset.

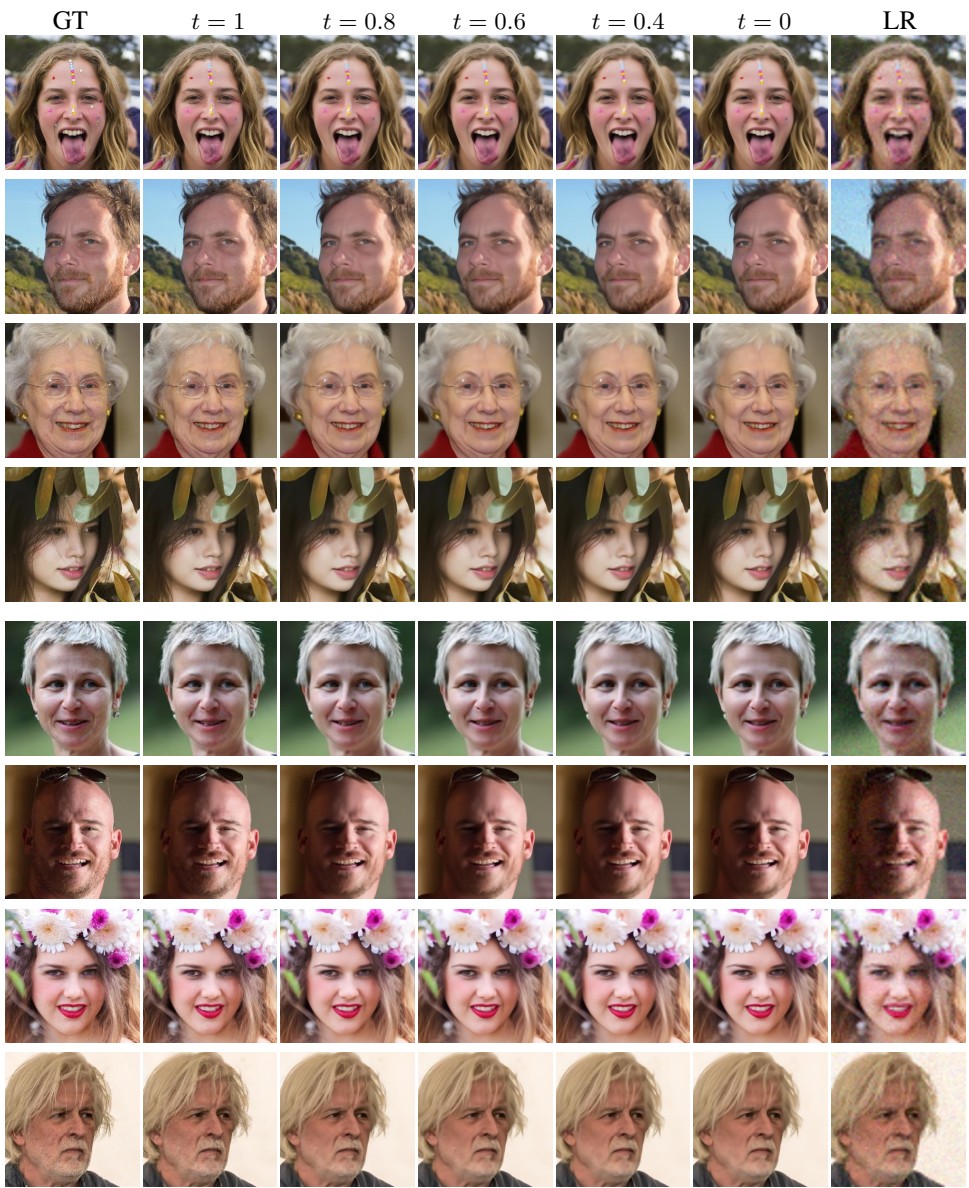

Figure 13: Qualitative results of one-step model with different tunable $t$.

Ground Truth  Measurement   DPS (1000)   DDNM (100)  DiffPIR (100)  SITCOM (20)   Ours (1)

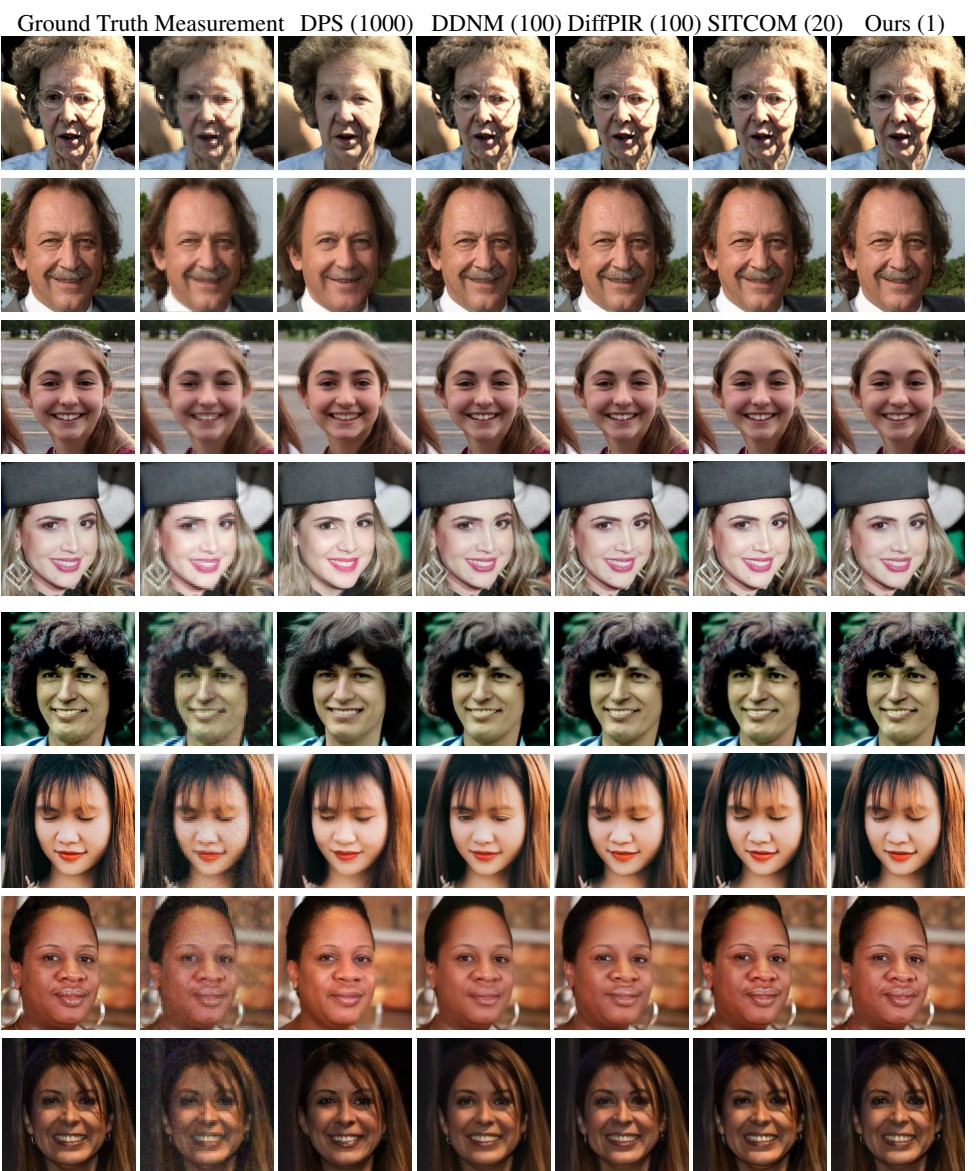

Figure 14: Qualitative comparisons on FFHQ dataset for $4\times$ SR with $\sigma_n = 0$ (first four rows) and $\sigma_n = 0.05$ (last four rows).

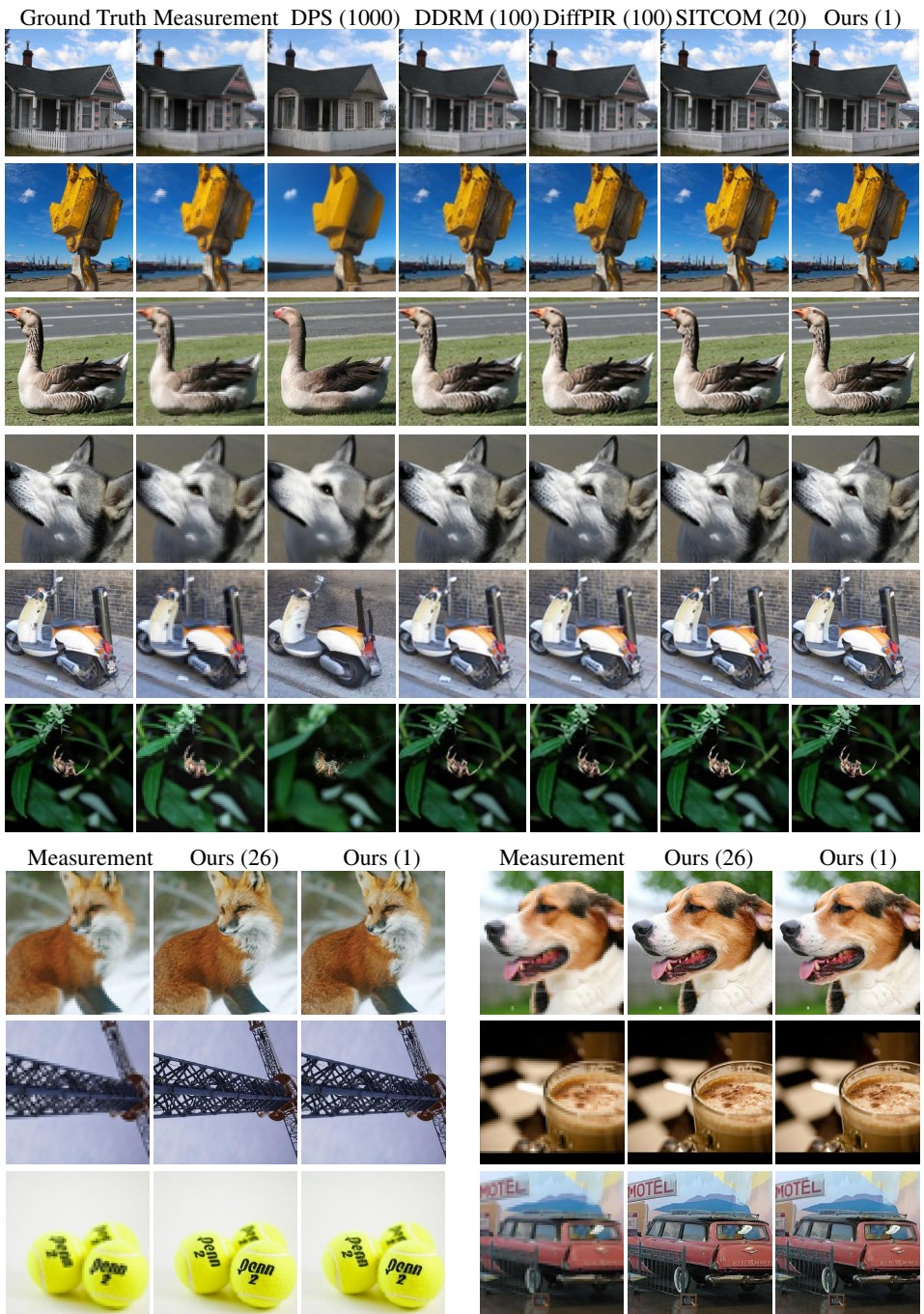

Figure 15: Qualitative comparisons on ImageNet dataset for noiseless $4\times$ SR.

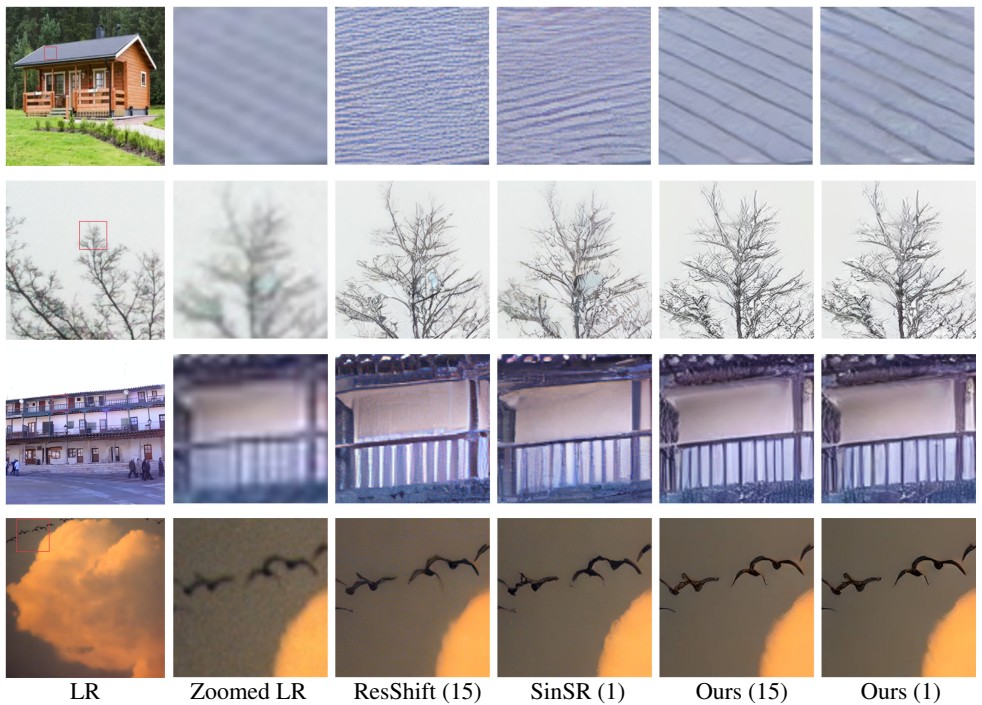

|       |            |              |           |           |          |
|-------|------------|--------------|-----------|-----------|----------|
| LR    | Zoomed LR  | ResShift (15)| SinSR (1) | Ours (15) | Ours (1) |

Figure 16: Qualitative comparisons for real-world 4× SR

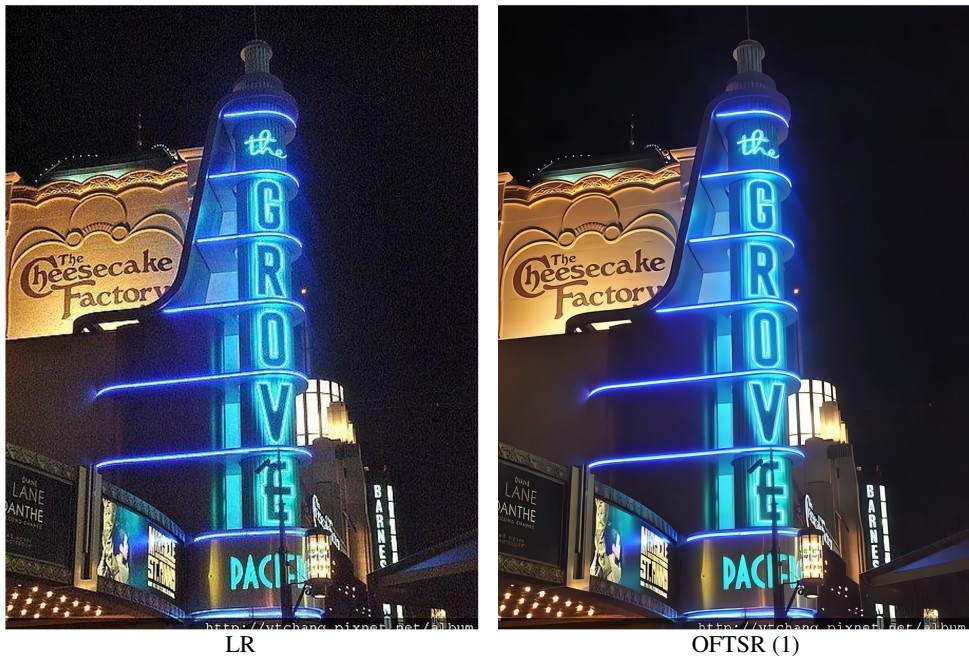

LR                                   OFTSR (1)

Figure 17: Visual result of restoring arbitrary scale LR image.

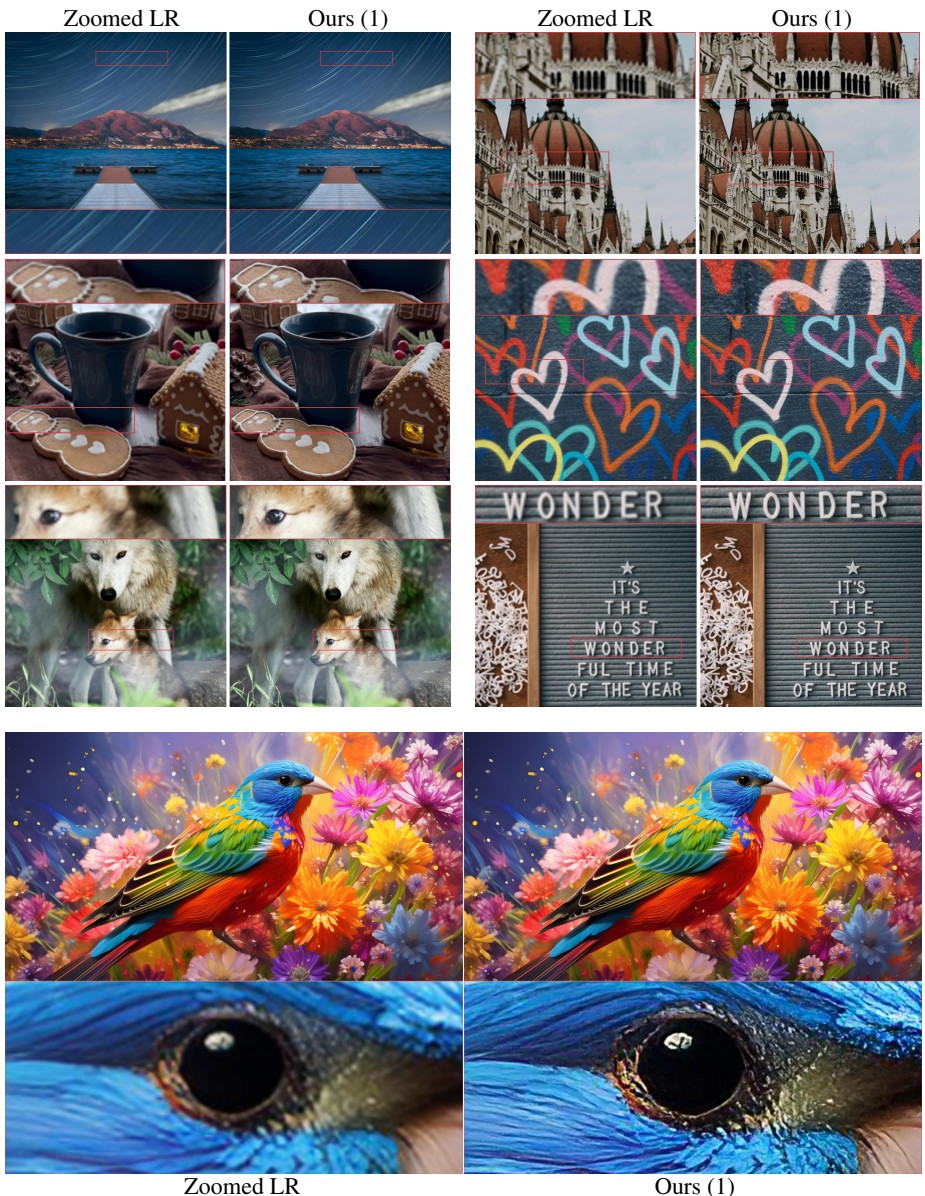

Figure 18: Qualitative results on real data and AI generated content using our $4\times$ SR model trained on DIV2K.

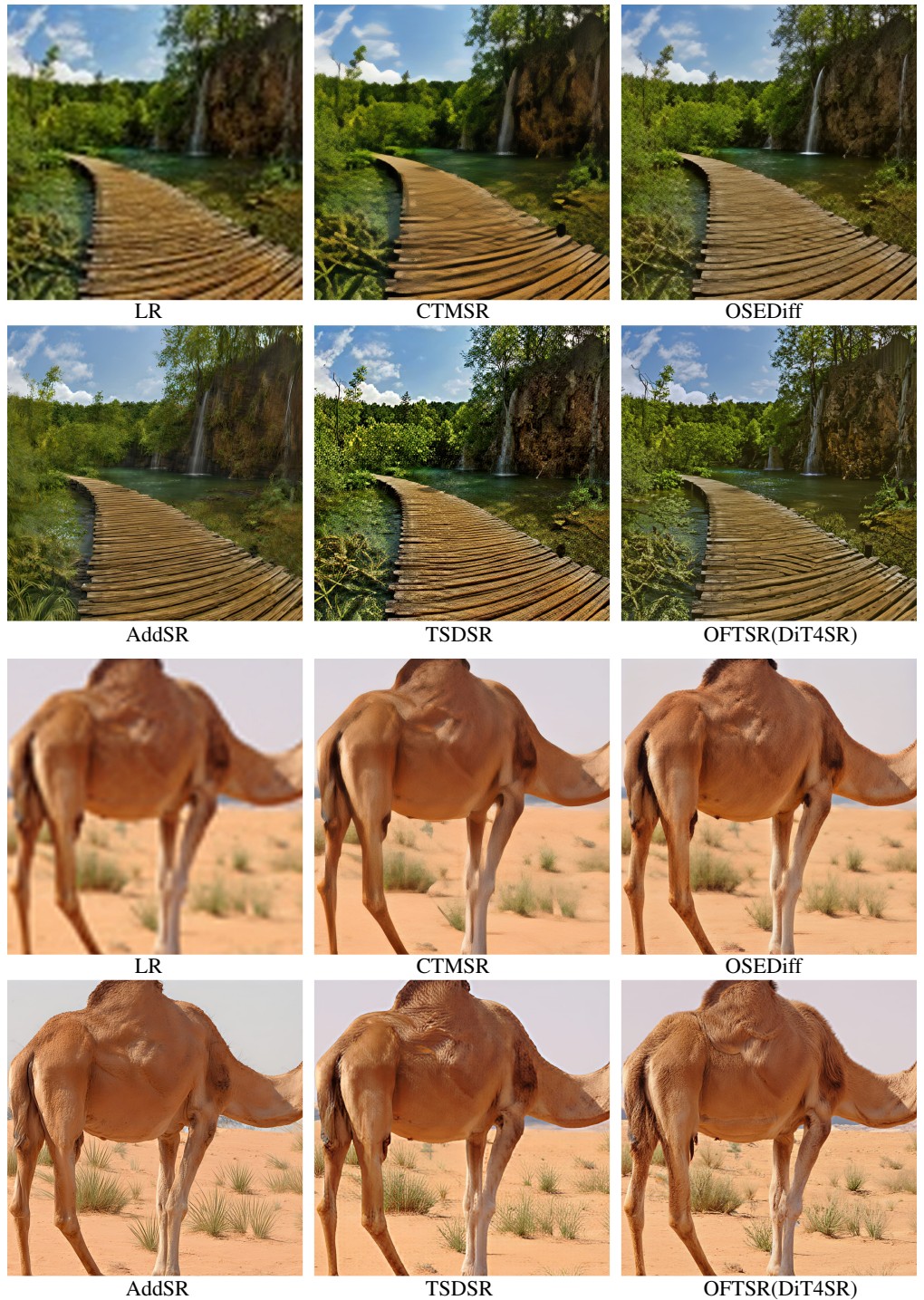

Figure 19: Qualitative comparisons for real-world 4× SR. OFTSR is distilled from DiT4SR. All methods perform 1 step inference.

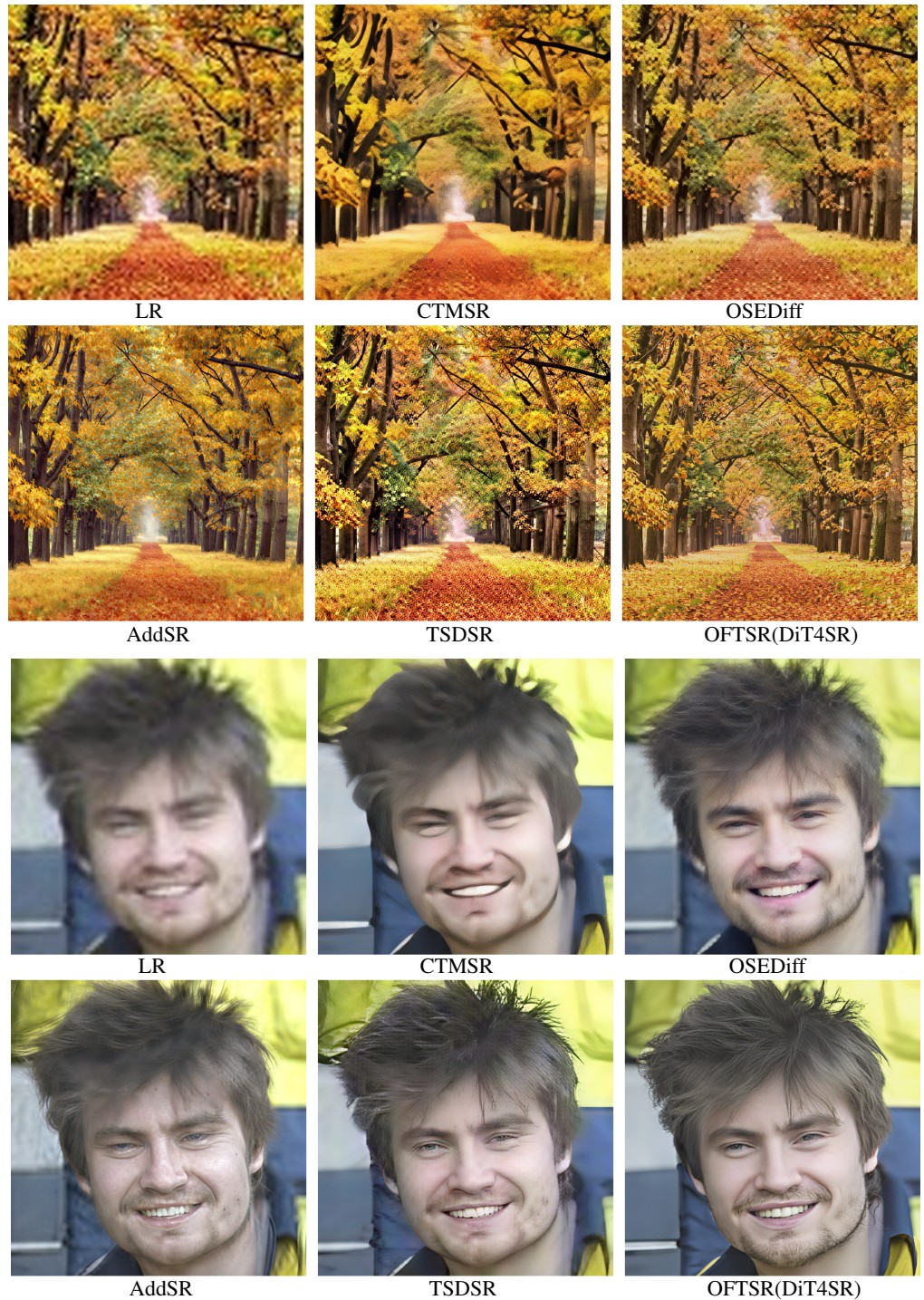

Figure 20: Qualitative comparisons for real-world 4× SR. OFTSR is distilled from DiT4SR. All methods perform 1 step inference.

