# OpenReview forum: "One-Step Flow for Image Super-Resolution with Tunable Fidelity-Realism Trade-offs"
_ICLR.cc/2026/Conference — ICLR 2026 Poster_

### Official Review · Reviewer_vEMX · 2025-10-30

**Soundness:** 3
**Presentation:** 3
**Contribution:** 2
**Rating:** 6
**Confidence:** 3

**Summary:**

The authors consider the super-resolution problem and propose a method to distill a pretrained flow model in a generator, which produces output on the PF-ODE trajectory for a given time “t” in 1 step. The ability to produce output at any “t” on the ODE trajectory provides a way to control the fidelity-realism trade-off, since for a bigger time “t”, samples are better in “realism” while for less time “t”, samples are better in “fidelity”. The authors evaluate their method on noisy and noiseless SR setups, and conduct some evaluations on real SR setups.

**Strengths:**

- The proposed method provides a clear and simple method to control the fidelity-realism tradeoff by choosing the time parameter “t” in the generator.
- The distilled model works in 1 step without significant degradation of the quality of generation.

**Weaknesses:**

- The proposed method is not theoretically novel in the sense that it is a special case of the general forward-distillation framework.
- Most of the evaluations and comparisons are done with simple downsampling or downsampling plus the addition of noise setups. For RealSR, only non-reference metrics are provided. For the ImageNet comparison, only one competitor distillation method (SinSR) is considered, while other competitive methods like OSEdiff are omitted.
- The authors compare with I2SB, but not with its various accelerated I2SB versions [1, 2, 3], which is strange since the proposed method is also distillation.

[1] Wang Y. et al. Implicit Image-to-Image Schrödinger Bridge for image restoration //Pattern Recognition. – 2025. – Т. 165. – С. 111627.

[2] He G. et al. Consistency diffusion bridge models //Advances in Neural Information Processing Systems. – 2024. – Т. 37. – С. 23516-23548.

[3] Gushchin N. et al. Inverse Bridge Matching Distillation //arXiv preprint arXiv:2502.01362. – 2025.

**Questions:**

Is there a way to automatically choose a time for the generator?

---

> ### Author Response · Authors · 2025-11-21
> **Response to Reviewer vEMX (1/2)**
>
> > [**W1**] The proposed method is not theoretically novel in the sense that it is a special case of the general forward-distillation framework.
>
> Our main contribution is a **principled distillation framework that enables a *continuous fidelity–realism trade-off* in a *single-step* SR model**, a capability not offered by existing distillation or consistency-based approaches.
>
> As clarified in Appendix B.2, forward distillation is concurrent work, our method serve as its discrete counterpart, and our formulation differs in several key aspects:
>
> - **Different objective and computational structure.**
>    Forward distillation is defined in *continuous time* and conceptually assumes the limit $dt \rightarrow 0$, requiring Jacobian evaluations similar to MeanFlow.
>    In contrast, our method is a **finite-step, discretized formulation** tailored specifically to the SR setting, avoiding Jacobians entirely and enabling efficient one-step inference.
>
> - **First application to SR and conditional flows with controllable trade-off.**
>    To our knowledge, we are the first to apply a forward-style trajectory-matching objective to **diffusion/flow-based SR models**, and to show that this leads to a **stable, interpretable fidelity–realism trade-off** controlled by a single parameter $t$.
>
>
> > [**W2**] Most of the evaluations and comparisons are done with simple downsampling or downsampling plus the addition of noise setups. For RealSR, only non-reference metrics are provided. For the ImageNet comparison, only one competitor distillation method (SinSR) is considered, while other competitive methods like OSEdiff are omitted.
>
>
>
> ### **(1) Real-world SR evaluation beyond simple degradations**
> We agree that real-world robustness is important.
> Our experiments already go beyond synthetic downsampling:
>
> - We follow the **established RealESRGAN degradation pipeline**, consistent with prior SR work.
> - We also evaluate on **RealSR**, a *genuinely real* dataset where even the ''GT'' images are real captures (and LR images are obtained by further ×4 downsampling). This setup is identical to recent works such as **TSDSR**, ensuring a fair and realistic evaluation protocol.
> - We have additionally included **reference-based metrics** for RealSR to complement non-reference metrics.
>
> These evaluations collectively demonstrate that OFTSR performs reliably under both synthetic and real-world degradations.
>
>
> ### **(2) Expanded comparisons on ImageNet-scale SR**
> For ImageNet-scale comparisons, we initially focused on **SinSR**, which is the most directly comparable one-step distillation baseline under the **same training resolution and conditions**.
>
> To address the reviewer's concern about broader coverage, we now include comparisons with **OSEDiff** (Table 6), which uses a SD-based **DiT4SR** $512\times512$ teacher.
> This provides a fair benchmark against recent SOTA diffusion-based SR methods.
> Together with our OFTSR(DiT4SR) results, these expanded baselines give a **more complete and up-to-date evaluation** of competitiveness across both standard and large-scale SR settings.
>
>
> ## RealLQ250 Dataset
>
> | Metric | SinSR-1s | CTMSR-1s | AddSR-1s | OSEDiff-1s | TSDSR-1s | **Ours (ResShift)-1s** | **Ours (DiT4SR)-1s** |
> |--------|----------|----------|----------|------------|----------|------------------------|---------------------|
> | NIQE ↓ | 5.8200 | 4.5835 | 4.9235 | 3.9656 | **3.4868** | 4.0731 | *3.7802* |
> | MUSIQ ↑ | 63.73 | 68.00 | 66.82 | 69.55 | *72.09* | 67.32 | **72.60** |
> | MANIQA ↑ | 0.5161 | 0.5078 | 0.5304 | 0.5782 | 0.5829 | 0.5287 | **0.5904** |
> | CLIPIQA ↑ | 0.6990 | 0.6706 | 0.6437 | 0.6725 | *0.7221* | 0.6532 | **0.7252** |
> | LIQE ↑ | 3.2578 | 3.3373 | 3.4929 | 3.9039 | *4.0834* | 3.7211 | **4.1122** |
>
> **Note:** ↑ indicates higher is better, ↓ indicates lower is better. **Bold** indicates best results, *italic* indicates second best.
>
> > Duan, Zheng-Peng, Jiawei Zhang, Xin Jin, Ziheng Zhang, Zheng Xiong, Dongqing Zou, Jimmy S. Ren, Chunle Guo, and Chongyi Li. "Dit4sr: Taming diffusion transformer for real-world image super-resolution." In Proceedings of the IEEE/CVF International Conference on Computer Vision, pp. 18948-18958. 2025.

---

> ### Author Response · Authors · 2025-11-21
> **Response to Reviewer vEMX (2/2)**
>
> > [**W3**] The authors compare with I2SB, but not with its various accelerated I2SB versions [1, 2, 3], which is strange since the proposed method is also distillation.
>
> We appreciate the suggestion to compare against accelerated I2SB variants.
> However, only **one** of the accelerated methods, **I3SB** [1], is publicly available.
> The other two approaches, **CDBM** [2] and **IBMD** [3], are **not open-sourced**, making reproducible comparison infeasible.
> To ensure experimental rigor, we include results for **I3SB** (in Table 3) and add a conceptual discussion of all three variants.
> We have cited these work and included the discussion in the appendix in the updated manuscript.
>
> ### **(1) On I3SB (Implicit Image-to-Image Schrödinger Bridge) [1]**
> I3SB provides an improved sampler for **pretrained I2SB models**, analogous to DDIM for DDPM.
> Although it improves inference speed, **I3SB still exhibits the same fundamental behavior** as I2SB:
>
> - **Many-step sampling** is required for high perceptual realism.
> - **1-step sampling** produces high fidelity but noticeably worse realism.
>
> In contrast, our distillation approach yields **substantially better realism at 1 NFE** while also providing a **continuous fidelity–realism trade-off**.
>
> ### **(2) On CDBM (Consistency diffusion bridge models) [2]**
> CDBM proposes consistency-style training and distillation for *Diffusion Bridge Models*.
> However, its experimental scope is **limited to small image-to-image tasks**, such as:
>
> - Edges → Handbags
> - DIODE-Outdoor
> - Image inpainting on ImageNet
>
> Since **no SR evaluation** is provided and no code is released, a direct comparison is not feasible.
>
>
> ### **(3) On IBMD (Inverse Bridge Matching Distillation) [3]**
> IBMD introduces a DMD-like distribution matching procedure for conditional bridge models.
> However:
>
> - It requires training an **additional auxiliary model $\phi$**, increasing computation and potentially affecting stability.
> - Published results show **1-step performance comparable to I2SB's 1000-step sampling**, suggesting no advantage over our distillation approach.
>
> Given these limitations and the lack of code, it is not feasible to compare IBMD empirically.
>
>
> > [**Q1**] Is there a way to automatically choose a time for the generator?
>
> In practice, the fidelity–realism parameter $t$ is not highly sensitive to the dataset or degradation type.
> Across all experiments, its effective range stays consistent, and the choice is naturally guided by user preference, $t\approx0$ favoring maximal fidelity and $t\approx1$ favoring maximal realism.
>
> If a specific target metric or domain preference is provided, $t$ can be selected automatically by optimizing $t$ on a small validation set to match the desired objective.

---

### Official Review · Reviewer_dEo3 · 2025-10-31

**Soundness:** 3
**Presentation:** 3
**Contribution:** 2
**Rating:** 4
**Confidence:** 3

**Summary:**

The paper introduces OFTSR, a novel one-step image super-resolution (SR) method using a flow-based framework that incorporates tunable fidelity-realism trade-offs. It leverages a teacher-student distillation approach, with a noise-augmented conditional flow model for SR. The student model is trained through a distillation strategy, aligning its predictions along the probability flow ordinary differential equation (PF-ODE) trajectory of the teacher. The method is demonstrated to achieve state-of-the-art performance for one-step SR on several datasets like FFHQ, DIV2K, and ImageNet. A unique feature of OFTSR is its ability to adjust the fidelity-realism trade-off via a single hyperparameter.

**Strengths:**

1. The proposed OFTSR integrates flow modeling with distillation, achieving high-quality super-resolution in a single inference step. It reduces inference cost by over an order of magnitude compared to diffusion-based methods, while maintaining comparable perceptual quality.

2. The approach’s ability to control the fidelity-realism trade-off through a single hyperparameter adds significant flexibility. OFTSR balances the perceptual quality and fidelity of SR results.

3. The training cost is less than previous methods, and the method seems to be a general and extensible solution for fast generative restoration.

**Weaknesses:**

1. The paper does not compare with recent state-of-the-art methods such as OSEDiff [1], AddSR [2], CTMSR[3], and TSDSR [4]. This omission makes it difficult to fully evaluate the competitiveness of the proposed method in the context of the latest advances in this field.

2. The experiments are mainly conducted at a resolution of 256×256, without investigating the model’s generalization capability at higher resolutions.

3. The paper lacks an ablation study on SFT a one-step model initialized from the teacher model. I suggest the authors include this experiment to better demonstrate the effectiveness of the proposed method.

[1] One-Step Effective Diffusion Network for Real-World Image Super-Resolution. NeurlPS2024

[2] AddSR: Accelerating Diffusion-based Blind Super-Resolution with Adversarial Diffusion Distillation.

[3] Consistency Trajectory Matching for One-Step Generative Super-Resolution. ICCV 2025

[4] TSD-SR: One-Step Diffusion with Target Score Distillation for Real-World Image Super-Resolution. CVPR 2025

**Questions:**

How sensitive is the fidelity–realism control parameter $t$ to different datasets or degradation types? Can it be automatically optimized for a given target domain?

---

If the above weaknesses and questions are addressed, I would consider increasing my rating.

---

> ### Author Response · Authors · 2025-11-21
> **Response to Reviewer dEo3 (1/2)**
>
> > [**W1**] The paper does not compare with recent state-of-the-art methods such as OSEDiff [1], AddSR [2], CTMSR[3], and TSDSR [4]. This omission makes it difficult to fully evaluate the competitiveness of the proposed method in the context of the latest advances in this field.
>
> We appreciate the suggestion to include these recent baselines.
> It is important to note that **OSEDiff, AddSR, CTMSR, and TSDSR** are all trained at **$512\times512$** resolution and rely on **Stable-Diffusion–based backbones**, which provide significantly stronger generative priors than the **$256\times256$** SR setting used in our main experiments. We have also updated our paper manuscript to make it more structured and clear.
>
> To ensure a fair comparison, we provide additional experiments by evaluating **OFTSR distilled from DiT4SR**, a large-scale SR model adapted from **SD3.5**, belonging to the same model family used by TSDSR and other SD-based methods.
> As reported in the following table and **Table 6** in the revised manuscript, **OFTSR (DiT4SR)** achieves performance competitive with these state-of-the-art approaches on both synthetic real SR (DIV2K-Val) and real-world SR (RealLQ250) benchmarks, demonstrating that our one-step distilled model remains strong even when applied to modern high-capacity backbones.
>
>
> ## DIV2K-Val Dataset
>
> | Metric | SinSR-1s | CTMSR-1s | AddSR-1s | OSEDiff-1s | TSDSR-1s | **Ours (ResShift)-1s** | **Ours (DiT4SR)-1s** |
> |--------|----------|----------|----------|------------|----------|------------------------|---------------------|
> | PSNR ↑ | *24.50* | **24.87** | 22.39 | 23.86 | 22.17 | 23.91 | 22.80 |
> | SSIM ↑ | 0.6136 | **0.6349** | 0.5652 | *0.6233* | 0.5680 | 0.6073 | 0.5774 |
> | LPIPS ↓ | 0.3164 | 0.3011 | 0.3728 | 0.2896 | **0.2679** | 0.3226 | *0.2716* |
> | DISTS ↓ | 0.2110 | 0.2102 | 0.2387 | 0.1999 | *0.1901* | 0.2081 | **0.1889** |
> | FID ↓ | 131.96 | 126.49 | 133.78 | 100.53 | *103.49* | 133.30 | **98.27** |
> | NIQE ↓ | 6.1721 | 5.3036 | 5.9929 | 4.9741 | **4.6621** | 4.9061 | *4.8399* |
> | MUSIQ ↑ | 64.26 | 66.59 | 63.39 | 68.53 | **71.19** | 68.71 | *70.25* |
> | MANIQA ↑ | 0.5442 | 0.5146 | 0.5657 | *0.6111* | 0.6010 | 0.5464 | **0.6145** |
> | CLIPIQA ↑ | 0.6687 | 0.6602 | 0.5734 | 0.6692 | *0.7221* | 0.6545 | **0.7233** |
>
> ## RealLQ250 Dataset
>
> | Metric | SinSR-1s | CTMSR-1s | AddSR-1s | OSEDiff-1s | TSDSR-1s | **Ours (ResShift)-1s** | **Ours (DiT4SR)-1s** |
> |--------|----------|----------|----------|------------|----------|------------------------|---------------------|
> | NIQE ↓ | 5.8200 | 4.5835 | 4.9235 | 3.9656 | **3.4868** | 4.0731 | *3.7802* |
> | MUSIQ ↑ | 63.73 | 68.00 | 66.82 | 69.55 | *72.09* | 67.32 | **72.60** |
> | MANIQA ↑ | 0.5161 | 0.5078 | 0.5304 | 0.5782 | 0.5829 | 0.5287 | **0.5904** |
> | CLIPIQA ↑ | 0.6990 | 0.6706 | 0.6437 | 0.6725 | *0.7221* | 0.6532 | **0.7252** |
> | LIQE ↑ | 3.2578 | 3.3373 | 3.4929 | 3.9039 | *4.0834* | 3.7211 | **4.1122** |
>
> **Note:** ↑ indicates higher is better, ↓ indicates lower is better. **Bold** indicates best results, *italic* indicates second best.
>
> > Duan, Zheng-Peng, Jiawei Zhang, Xin Jin, Ziheng Zhang, Zheng Xiong, Dongqing Zou, Jimmy S. Ren, Chunle Guo, and Chongyi Li. "Dit4sr: Taming diffusion transformer for real-world image super-resolution." In Proceedings of the IEEE/CVF International Conference on Computer Vision, pp. 18948-18958. 2025.

---

> ### Author Response · Authors · 2025-11-21
> **Response to Reviewer dEo3 (2/2)**
>
> > [**W2**] The experiments are mainly conducted at a resolution of 256×256, without investigating the model's generalization capability at higher resolutions.
>
>
> ### **(1) Strong high-resolution generalization without retraining**
> Although our main training resolution is $256\times256$, our experiments already demonstrate that the model **generalizes well to higher resolutions**:
>
> - **RealSet80** contains images with *varied and often higher native resolutions*, and our $64\rightarrow256$ model performs robustly without modification (see Table 5).
> - In **RealLQ 4×**, we upscale **$128 \rightarrow 512$** using a model trained *only* on **ImageNet $64\rightarrow256$** (see Table 5, 6).
> - In Appendix G.3, we also provided exprimental results on different scaling factors (8×) showing the model's flexibility.
> - **Figure 16** further presents **$512\times512$** reconstructions, while **Figure 17** shows **arbitrary-scale SR** without retraining.
>
> These results confirm that OFTSR inherits the teacher's architectural scalability and handles high-resolution inference effectively.
>
> ### **(2) Training directly at higher resolutions**
> Beyond generalization, we also demonstrate that OFTSR can be **trained natively at higher resolutions**:
>
> - **OFTSR (DiT4SR)** operates at **$512\times512$**, using the DIT4SR teacher adapted from modern SD.
>
>
> > [**W3**] The paper lacks an ablation study on SFT a one-step model initialized from the teacher model. I suggest the authors include this experiment to better demonstrate the effectiveness of the proposed method.
>
>
> We appreciate the suggestion, but we believe the intent of ''SFT'' here is unclear.
> (1) If the reviewer means SFT of a one-step student using ground-truth HR images, then this is not comparable to our setting, since our student never sees GT during training.
> (2) If instead SFT means fine-tuning the student using the teacher's original pre-training objective, then this does not yield a one-step generator, the student would still require many NFEs.
> Our distillation objective is specifically designed to transfer the teacher's multi-step ODE behavior into a single forward pass, which standard SFT cannot achieve.
> We will clarify this distinction in the paper, and we welcome further details from the reviewer if they had a specific SFT variant in mind.
>
>
> > [**Q1**] How sensitive is the fidelity–realism control parameter $t$ to different datasets or degradation types? Can it be automatically optimized for a given target domain?
>
> In practice, the fidelity–realism parameter $t$ is not highly sensitive to the dataset or degradation type.
> Across all experiments, its effective range stays consistent, and the choice is naturally guided by user preference, $t\approx0$ favoring maximal fidelity and $t\approx1$ favoring maximal realism.
>
> If a specific target metric or domain preference is provided, $t$ can be selected automatically by optimizing $t$ on a small validation set to match the desired objective.

---

> ### Comment · Reviewer_dEo3 · 2025-11-28
>
> Thank you for the detailed explanations and additional experimental results. I believe the authors have satisfactorily resolved the key issues raised in my earlier review. After careful consideration, I am now inclined to increase my score to 6.

---

> > ### Author Response · Authors · 2025-11-28
> >
> > Thank you very much for your thoughtful follow-up. We truly appreciate the time and effort you put into reviewing and the constructive feedback.

---

### Official Review · Reviewer_U3BC · 2025-10-31

**Soundness:** 2
**Presentation:** 3
**Contribution:** 2
**Rating:** 4
**Confidence:** 4

**Summary:**

This paper introduces OFTSR, a flow-based one-step super-resolution (SR) framework that builds upon rectified flow models and distillation. The key idea is to train a conditional flow-based SR teacher and distill it into a single-step student whose predictions are constrained to lie on the same probability flow ODE trajectory as the teacher. This constraint allows the one-step model to maintain a tunable fidelity–realism trade-off controlled by a continuous parameter (t). Experiments on FFHQ, DIV2K, and ImageNet show that OFTSR achieves competitive or state-of-the-art performance among one-step methods, while being more efficient than multi-step diffusion-based baselines.

I found this to be a well-executed and empirically strong paper, but the conceptual innovation is somewhat limited. The proposed ODE alignment distillation is a reasonable technical improvement, yet it does not introduce a fundamentally new idea beyond existing consistency and distillation methods. The experimental section is solid, but the lack of theoretical depth and real-world validation makes it fall slightly short of the bar for acceptance. That said, the work is promising and practical, and I could imagine it being accepted after stronger justification of its theoretical grounding and broader validation.

**Strengths:**

- The paper’s formulation is clear, the equations are consistent with prior work in diffusion and flow-based modeling, and the proposed ODE alignment loss is a neat way to adapt teacher–student distillation for conditional SR.
- The experimental section is impressive in scope. The authors benchmark across three datasets, consider both noisy and noiseless conditions, include training-free and training-based baselines, and conduct detailed ablations on hyperparameters and solvers.
- The method achieves true one-step inference while preserving control over perceptual vs. fidelity quality, which is a meaningful step forward for efficient SR.
- The paper is well-written and easy to follow, with intuitive figures that make the underlying intuition accessible.

**Weaknesses:**

- Conceptually, the work is an incremental extension of existing distillation or consistency-based methods (e.g., BOOT, MeanFlow, Consistency Models). The ODE-alignment constraint is interesting but feels more like a technical refinement than a new paradigm. This limits the paper’s theoretical depth.
- The ODE alignment loss is mostly empirically motivated. There’s no clear analysis of when or why this constraint ensures better one-step consistency, nor how it relates formally to the underlying PF-ODE or optimal transport formulations.
- Although the experiments are extensive, most are performed under controlled synthetic conditions. The “real-world” SR experiments use degradations simulated with RealESRGAN, which doesn’t fully validate real-world robustness. I would have liked to see more genuinely real test data or domain-shift scenarios.
- All baselines are diffusion/flow-based. The paper does not compare against any deterministic SR networks (e.g., SwinIR, NAFNet), which are used as efficiency benchmark in practice.
- The paper doesn’t discuss whether OFTSR maintains diversity in outputs (e.g., multiple plausible HR reconstructions per LR input) or generalizes across scales/resolutions. These are important in assessing flow-based generative SR methods.
- While the interpolation parameter t is central to the paper, there’s little analysis of its sensitivity or failure modes. Some visualizations (e.g., at extreme t) would help validate robustness.

**Questions:**

1) How sensitive is OFTSR to the specific choice of teacher (e.g., if trained with different flow objectives)?
2) Can your ODE alignment loss be extended to a few-step regime (e.g., 2–4 NFEs)?
3) Is the model capable of diverse sampling (multiple HRs per LR) or is it deterministic?
4) How does performance compare to lightweight CNN/transformer SR models under equal runtime?
5) Could you provide visual examples showing failure or instability at extreme t values?

---

> ### Author Response · Authors · 2025-11-21
> **Response to Reviewer U3BC (1/3)**
>
> > [**W1**] Conceptually, the work is an incremental extension of existing distillation or consistency-based methods (e.g., BOOT, MeanFlow, Consistency Models). The ODE-alignment constraint is interesting but feels more like a technical refinement than a new paradigm. This limits the paper's theoretical depth.
>
> We respectfully disagree that our method is an incremental refinement of BOOT, or consistency models. While these approaches align trajectories or enforce consistency, **none of them address the central problem tackled in our work: enabling a controllable fidelity–realism trade-off in a one-step SR model**.
> Our main contribution is a principled distillation framework that enables a continuous fidelity–realism trade-off for single-step SR models, which is a novel capability not provided by prior methods.
>
> > [**W2**] The ODE alignment loss is mostly empirically motivated. There's no clear analysis of when or why this constraint ensures better one-step consistency, nor how it relates formally to the underlying PF-ODE or optimal transport formulations.
>
> Intuitively, the alignment loss is motivated by aligning the student's velocity field with the teacher's along the trajectory.
> While our ODE-alignment loss is empirically motivated, it is a natural extension of boundary losses.  Empirically, we find that this alignment improves one-step SR quality while preserving controllability over the trade-off.
>
> > [**W3 & W4 & Q4**] [*W3*] Although the experiments are extensive, most are performed under controlled synthetic conditions. The ''real-world'' SR experiments use degradations simulated with RealESRGAN, which doesn't fully validate real-world robustness. I would have liked to see more genuinely real test data or domain-shift scenarios.
> [*W4*] All baselines are diffusion/flow-based. The paper does not compare against any deterministic SR networks (e.g., SwinIR, NAFNet), which are used as efficiency benchmark in practice.
> [*Q4*] How does performance compare to lightweight CNN/transformer SR models under equal runtime?
>
> We thank the reviewer for these valuable suggestions, and we have addressed them by adding new experiments and analyses in the revised manuscript.
>
> ### **(1) Real-world evaluation beyond synthetic degradations**
> We agree that robustness on genuinely real images is essential.
> In addition to the RealESRGAN-based synthetic setting, our experiments already include **two real-world datasets** (Table 5, 6 in the updated manuscript):
> - **RealSet80** (from ResShift)
> - **RealLQ250** (from DreamClear [1])
>
> both of which contain *naturally degraded*, in-the-wild images and do **not** rely on the RealESRGAN degradation pipeline.
> Furthermore, **Figure 16** presents qualitative results on genuinely real **128 → 512** SR, demonstrating flexibility of our method.
> These evaluations show that our method remains stable under realistic conditions.
>
>
> ### **(2) Comparison with deterministic SR networks**
> We have added comparisons with **SwinIR** and **NAFNet** in the following table and **Table 5** of the revised manuscript.
> The results show that our distilled models outperform these deterministic baselines on multiple perceptual quality metrics (NIQE, MUSIQ, MANIQA, CLIPIQA, LIQE) while maintaining competitive efficiency (one-step inference).
> Regarding the runtime, our one-step model runs in comparable time to SwinIR and NAFNet, as all three methods require only a single forward pass through a neural network. The model sizes of diffusion/flow-based methods are generally larger due to their generative nature.
>
> | Datasets | Method | NIQE ↓ | MUSIQ ↑ | MANIQA ↑ | CLIPIQA ↑ | LIQE ↑ |
> |----------|--------|--------|---------|----------|-----------|--------|
> | **RealSet80** | SwinIR | **4.1601** | 63.72 | 0.5444 | 0.5919 | 3.6479 |
> | | NAFNet | 8.8794 | 35.16 | 0.3975 | 0.5289 | 1.0969 |
> | | ResShift-15s | 6.1955 | 61.35 | 0.5318 | 0.6702 | 3.4473 |
> | | SinSR-1s | 5.6182 | 63.96 | 0.5376 | **0.7242** | 3.6072 |
> | | **Ours-15s** | *4.3713* | *66.90* | **0.5617** | 0.6797 | *3.9982* |
> | | **Ours distilled-1s (t=1)** | *4.1826* | **67.46** | *0.5570* | *0.6904* | **4.0168** |
> | **RealLQ250** | SwinIR | 4.1628 | 60.48 | 0.5104 | 0.5352 | 3.0883 |
> | | NAFNet | 9.5524 | 25.97 | 0.3360 | 0.4095 | 1.0512 |
> | | ResShift-15s | 6.5731 | 59.98 | 0.5003 | 0.6239 | 2.9340 |
> | | SinSR-1s | 5.8200 | 63.73 | 0.5161 | **0.6990** | 3.2578 |
> | | **Ours-15s** | 4.2848 | *67.15* | **0.5481** | 0.6520 | **3.8367** |
> | | **Ours distilled-1s (t=1)** | **4.0731** | **67.32** | *0.5287* | *0.6532* | *3.7211* |
>
> **Note:** ↑ indicates higher is better, ↓ indicates lower is better. **Bold** indicates best results, *italic* indicates second best.
>
> > [1] Ai, Yuang, Xiaoqiang Zhou, Huaibo Huang, Xiaotian Han, Zhengyu Chen, Quanzeng You, and Hongxia Yang. "DreamClear: High-Capacity Real-World Image Restoration with Privacy-Safe Dataset Curation." Advances in Neural Information Processing Systems 37 (2024): 55443-55469.

---

> ### Author Response · Authors · 2025-11-21
> **Response to Reviewer U3BC (2/3)**
>
> > [**W5 & Q3**] [*W5*] The paper doesn't discuss whether OFTSR maintains diversity in outputs (e.g., multiple plausible HR reconstructions per LR input) or generalizes across scales/resolutions. These are important in assessing flow-based generative SR methods.
> [*Q3*] Is the model capable of diverse sampling (multiple HRs per LR) or is it deterministic?
>
>
> ### **(1) Output diversity**
> OFTSR **is not deterministic**.
> As described in Sec. 3.1, the **noise-augmented initialization** injects stochasticity into the inference process.
> This enables the model to generate **multiple plausible HR reconstructions** for the same LR input.
> We provide diverse-sampling examples in Appendix Fig. 11, which visually demonstrate this variability.
> Moreover, all teacher models used in our work support diverse SR, and the distilled one-step student naturally inherits this property.
>
>
> ### **(2) Cross-scale and cross-resolution generalization**
>
> Our method also **generalizes across scales and resolutions**:
> - **RealSet80** contains LR images with *non-uniform and unknown* native scales; OFTSR performs robustly on this dataset without retraining.
> - In the **RealLQ 4×** experiments, we upscale **128 → 512** using a model trained only on **ImageNet 64 → 256**, demonstrating strong resolution generalization.
> - The visualization of arbitary resolution SR is shown in **Figure 17**.
> - In Appendix G.3, we also provided exprimental results on different scaling factors (8×) showing the model's flexibility.
> - Because the one-step model preserves the teacher's architectural structure, a teacher such as **ResShift (Swin-transformer backbone)** allows the student to handle resolutions *larger than those seen during training*.
>
>
> [**W6 & Q5**] [*W6*] While the interpolation parameter t is central to the paper, there's little analysis of its sensitivity or failure modes. Some visualizations (e.g., at extreme t) would help validate robustness.
> [*Q5*] Could you provide visual examples showing failure or instability at extreme t values?
>
> ### **(1) Behavior across the full valid range $t \in [0,1]$**
> We appreciate the reviewer's concern regarding sensitivity and potential failure modes.
> As shown in **Fig. 13**, varying $t$ across its entire valid range produces **stable and predictable behavior** along the realism–fidelity continuum. We do not observe artifacts, instabilities, or mode collapse at the endpoints $t = 0$ or $t = 1$.
>
> ### **(2) Behavior outside the valid range**
> To further address the reviewer's question, we include **OOD-$t$** visualizations in Appendix **Fig. 10**.
> When using invalid values such as $t = 2$ or $t = -0.5$, the model exhibits noticeable degradation, confirming that the method is naturally defined on the interval $[0,1]$.
> These examples help illustrate that the trade-off mechanism is stable within the intended range but not meaningful outside it.

---

> ### Author Response · Authors · 2025-11-21
> **Response to Reviewer U3BC (3/3)**
>
> > [**Q1**] How sensitive is OFTSR to the specific choice of teacher (e.g., if trained with different flow objectives)?
>
> OFTSR distillation is designed to be **teacher-agnostic**, and our experiments confirm strong robustness across different teacher architectures and flow objectives. We successfully distill:
>
> - an **off-the-shelf ResShift** model (latent-space flow),
> - our own **noise-augmented conditional-flow** teacher (pixel-space flow and latent-space flow) on several datasets, and
> - a **large-scale SD-based DiT4SR** model (rectified-flow).
>
> The consistency of results across these diverse teachers demonstrates that OFTSR is **not sensitive** to the specific flow objective used by the teacher.
>
>
> > [**Q2**] Can your ODE alignment loss be extended to a few-step regime (e.g., 2–4 NFEs)?
>
> In this work we mainly focus on the strict one-step regime: our analysis are tailored to a single NFE student. Extending the same idea to few-step samplers (2–4 NFEs) would require redesigning the student model and training objective, which is beyond our current scope.
> We consider developing multi-step ODE-aligned students as a promising direction for future work.

---

### Official Review · Reviewer_835P · 2025-11-01

**Soundness:** 3
**Presentation:** 3
**Contribution:** 3
**Rating:** 8
**Confidence:** 3

**Summary:**

This paper introduces OFTSR, a one-step image super-resolution method that preserves flexible control over the fidelity-realism trade-off. The approach uses a two-stage pipeline: first, training a noise-augmented conditional rectified flow model (teacher) that expands the initial distribution by adding Gaussian noise to low-resolution images, then distilling it into a one-step model (student) by constraining predictions to lie on the same ODE trajectory as the teacher. At inference, users can adjust a single parameter t to generate outputs ranging from high-fidelity/blurry (t=0) to high-realism/sharp (t=1) in just one forward pass, achieving state-of-the-art performance on FFHQ, DIV2K, and ImageNet datasets while being significantly more efficient than multi-step diffusion methods.

**Strengths:**

The paper's main strength is achieving an unprecedented combination of efficiency and flexibility, delivering tunable fidelity-realism trade-offs in a single forward pass, which no prior one-step method accomplishes. The technical approach is elegant and well-motivated: the noise-augmented conditioning prevents mode collapse while the ODE-trajectory constraint ensures the distilled model inherits the teacher's trade-off properties. The results achieved state-of-the-art performance across multiple benchmarks (FFHQ, DIV2K, ImageNet) while requiring only 1 NFE, making it faster at inference. The method is also applied to different teacher models and various tasks (noiseless SR, noisy SR, real-world SR, and different scale factors). It provides practical value, as a single trained model can serve multiple use cases by simply adjusting the t parameter.

**Weaknesses:**

The primary weakness is that the distilled model doesn't perfectly preserve the teacher's perception-distortion frontier. As shown in Figure 7, there's a slight shift in the trade-off curve, meaning the same t value doesn't yield identical fidelity-realism balances between teacher and student, likely due to the discrete step size (dt) approximation. The method's performance is fundamentally constrained by the teacher model's capabilities, and the two-stage training adds complexity and computational cost (though it remains more efficient than some alternatives, such as SinSR). The noise augmentation strength is a critical hyperparameter that requires careful tuning per task. The paper lacks a solid theoretical justification for why the proposed distillation loss preserves the trade-off properties or for what guarantees exist. Finally, some design choices (like dt=0.05) appear empirically driven rather than principled, and the ablation studies, while thorough, reveal sensitivity to these hyperparameters.

**Questions:**

LPIPS is a perceptual fidelity metric, while PSNR is a fidelity metric. In Figure 3, LPIPS and PNSR were used to demonstrate the transition between image realism and fidelity. A realism metric is preferred, e.g., FID, NIQE, etc. Again, as such a trade-off is a selling point, it would be better in the tables (1, 2, 3, 4). The metrics could be organized, e.g., realism metrics are put on the left and fidelity metrics on the right. That will help understand the differences against t values.

The SR baseline is established on diffusion and flow-based methods. Thus, the significance of the proposed method for general SR applications is unclear.

**Details Of Ethics Concerns:**

NIL

---

> ### Author Response · Authors · 2025-11-21
> **Response to Reviewer 835P (1/3)**
>
> > [**W1**] The primary weakness is that the distilled model doesn't perfectly preserve the teacher's perception-distortion frontier. As shown in Figure 7, there's a slight shift in the trade-off curve, meaning the same t value doesn't yield identical fidelity-realism balances between teacher and student, likely due to the discrete step size (dt) approximation.
>
> We acknowledge the reviewer's observation that the student's perception–distortion curve exhibits a slight shift relative to the teacher. This behavior is expected due to the discretization error introduced by collapsing the continuous trajectory into a single step. Importantly, the deviation is small and does not affect the core property we aim to preserve: **the student still supports a smooth and continuous fidelity–realism trade-off controlled by the same hyperparameter $t$**.
>
>
> > [**W2**] The method's performance is fundamentally constrained by the teacher model's capabilities, and the two-stage training adds complexity and computational cost (though it remains more efficient than some alternatives, such as SinSR).
>
>
> ### **(1) Teacher dependence is inherent but acknowledged**
> We agree that under purely teacher-supervised distillation the student cannot surpass the teacher's intrinsic capability.
> This limitation is explicitly discussed in Appendix Section C.
> Future extensions incorporating **ground-truth supervision such as GAN-style objectives**, or **reward models** could allow the student to exceed the teacher's performance.
>
>
> ### **(2) Two-stage pipeline is decoupled and flexible**
> Although our method uses a two-stage workflow, we argue that the stages are **fully decoupled**:
>
> - Our fisrt stage introduce a **general noise-augmented conditional flow** that can benefit similar SR models or conditional generation tasks.
> - The **distillation stage** can be applied to **any pretrained teacher**. Beyond our pre-trained flow models from the first stage, as shown in Tables 9 and 6 (in the updated manuscript), we successfully distill both:
>   - an **off-the-shelf ResShift model** for fair comparison with prior work.
>   - an **SD-based large model DIT4SR** to show the scalability of our distillation algorithm.
>
> > Duan, Zheng-Peng, Jiawei Zhang, Xin Jin, Ziheng Zhang, Zheng Xiong, Dongqing Zou, Jimmy S. Ren, Chunle Guo, and Chongyi Li. "Dit4sr: Taming diffusion transformer for real-world image super-resolution." In Proceedings of the IEEE/CVF International Conference on Computer Vision, pp. 18948-18958. 2025.

---

> ### Author Response · Authors · 2025-11-21
> **Response to Reviewer 835P (2/3)**
>
> > [**W3**] The noise augmentation strength is a critical hyperparameter that requires careful tuning per task.
>
> While the noise-augmentation strength is an important hyperparameter in the first stage, we find it **neither sensitive nor difficult to tune in practice**. Across all SR settings we tested, the optimal $\sigma_p$ consistently lies within a **stable and predictable range**. The choice is also guided by clear intuition:
>
> - **Stronger degradations** (e.g., Real-ESRGAN) benefit from **larger noise** ($\sigma_p \approx 0.5$).
> - **Cleaner ×4 SR settings** favor **smaller noise** ($\sigma_p \approx 0.1$).
>
>
>
> > [**W4**] The paper lacks a solid theoretical justification for why the proposed distillation loss preserves the trade-off properties or for what guarantees exist.
>
> Our distillation loss is derived directly from the teacher's noise-augmented flow ODE (Eq. 6) by matching the **Euler-step velocity field** along the same trajectory. This ensures that the student approximates the teacher's underlying vector field: the very mechanism that induces the teacher's realism–fidelity trade-off.
> For deeper theoretical insights, we look forward to future analyses building on recent works studying forward distillation.
>
>
> > [**W5**] Finally, some design choices (like dt=0.05) appear empirically driven rather than principled, and the ablation studies, while thorough, reveal sensitivity to these hyperparameters.
>
> We acknowledge that certain hyperparameters, such as $dt$, are chosen empirically. However, our ablations show that performance is **not strongly sensitive** to this choice: values in the range $[0.01-0.1]$ produce nearly identical outcomes. We have clarified this robustness in the paper and consider developing more principled guidance for these hyperparameters as a promising direction for future work.

---

> ### Author Response · Authors · 2025-11-21
> **Response to Reviewer 835P (3/3)**
>
> > [**Q1**] LPIPS is a perceptual fidelity metric, while PSNR is a fidelity metric. In Figure 3, LPIPS and PNSR were used to demonstrate the transition between image realism and fidelity. A realism metric is preferred, e.g., FID, NIQE, etc. Again, as such a trade-off is a selling point, it would be better in the tables (1, 2, 3, 4). The metrics could be organized, e.g., realism metrics are put on the left and fidelity metrics on the right. That will help understand the differences against t values.
>
> ### **(1) LPIPS is correlated with perceptual realism**
> We agree that realism-oriented metrics such as **FID** or **NIQE** are ideal for evaluating perceptual realism.
> However, Figure 3 plots *per-image* trade-off curves, and FID is not defined at the image level.
> LPIPS, while commonly viewed as a perceptual fidelity metric, is also known to **correlate strongly with perceptual realism**, which is why it was used there.
>
> To further validate the trend, we additionally computed **NIQE** across varying $t$:
>
> $$
> 7.930 (t=0),\ 7.869 (t=0.2),\ 7.819 (t=0.4),\
> 7.766 (t=0.6),\ 7.640 (t=0.8),\ 7.635 (t=1).
> $$
>
> NIQE improves monotonically as $t$ increases, consistent with LPIPS, confirming that the trade-off behavior is **robust across both realism and perceptual metrics**.
>
> ### **(2) Reorganizing metric tables for clarity**
> We appreciate the reviewer's suggestion and have updated Tables to make sure the fidelity metrics are on the left and use underscores to separate the two groups of metrics.
>
> > [**Q2**] The SR baseline is established on diffusion and flow-based methods. Thus, the significance of the proposed method for general SR applications is unclear.
>
> We appreciate the reviewer's question, though we are unsure what is meant by ''general SR applications,'' since **diffusion- and flow-based models currently constitute the state-of-the-art backbone** for modern SR systems. Our method is designed precisely for this family of models.
> If the reviewer has additional application scenarios or model classes in mind, we would welcome clarification so that we can address them more directly in the revision.

---

### Official Review · Reviewer_cMY2 · 2025-11-01

**Soundness:** 3
**Presentation:** 2
**Contribution:** 2
**Rating:** 4
**Confidence:** 4

**Summary:**

The paper introduces OFTSR, a conditional rectified flow framework for one-step image super-resolution that allows continuous control between fidelity and realism. The method consists of two stages: a noise-augmented conditional flow trained with a Variance-Preserving (VP) perturbation of the low-resolution (LR) input to enhance diversity and coverage of the HR distribution, and a distillation stage that aligns the student's single-step predictions with the teacher's probability flow ODE trajectory using a combination of distillation, alignment, and boundary losses. OFTSR demonstrates flexible, single-step inference and achieves strong quantitative and qualitative results on diverse datasets showing competitive or superior performance to baselines with significant gains in efficiency.

**Strengths:**

1. Tunable one step SR with explicit control.
The paper introduces an interpretable t-controlled mechanism that enables explicit adjustment of the fidelity-realism trade-off along the teacher’s ODE trajectory.


2. Efficiency and applicability.
The method requires only a single forward pass at inference while retaining flexibility typically reserved for multi-step models.

3. Strong empirical performance.
Evaluations span three datasets (DIV2K, FFHQ, ImageNet) and include noisy, noiseless, and real-world SR scenarios. The ablation studies (Tables 5-6) systematically analyze the effects of perturbation strength, solvers, and loss weights, providing solid empirical justification.

4. Clear Methodological Formulation.
The paper's derivation of the distillation and alignment losses (Equations 9-12, Algorithm 1) is mathematically consistent and clearly connected to existing distillation methods such as BOOT and forward distillation, offering both clarity and theoretical grounding.

**Weaknesses:**

1. Boundary of Novelty Relative to Existing Frameworks.
While the paper combines concepts appropriately, its originality is somewhat incremental.
The noise-augmented LR conditioning interpolates between SR3 and InDI as acknowledged in Section 3.1.
Similarly, the distillation loss in Eq. (9) is a constrained variant of forward distillation (Appendix B.2),
and the core idea of aligning student and teacher ODE trajectories resembles BOOT.
The contribution would be clearer if the authors better emphasized what properties uniquely arise from this specific combination in SR tasks.


2. Teacher dependence is acknowledged but under explored empirically.
Section C mentions that the student's performance is constrained by the teacher's capability,
but the experiments (Table 7) explore only a single ResShift teacher.
The lack of systematic testing across multiple teachers limits understanding of the method’s robustness
when teacher quality varies or exhibits bias.


3. Comparison with one step SR can be expanded.
The experimental comparison omits some recent one-step SR approaches [1, 2],
which could contextualize the claimed state-of-the-art results. Including these would strengthen the empirical claims.


4. Editorial issues. Minor editorial redundancies exist, such as repeated inference equations 13-14.

[Reference]

[1] TSD-SR: One-Step Diffusion with Target Score Distillation for Real-World Image Super-Resolution, CVPR 2025

[2] One-Step Effective Diffusion Network for Real-World Image Super-Resolution, NeurIPS 2024

**Questions:**

1. Clarification on Table 7: Could the authors explain the distinction between "OFTSR (ResShift teacher)" and "OFTSR (pre-train + distill)"?

2. Clarification on Table 8:
Two OFTSR rows are listed with identical NFEs but differing parameter counts (118.6M+55.3M vs 552.8M). Do these correspond to different architectures (e.g., ResShift-based vs standalone OFTSR) or to inclusion/exclusion of VAE components?

---

> ### Author Response · Authors · 2025-11-21
> **Response to Reviewer cMY2 (1/3)**
>
> > [**W1**] Boundary of Novelty Relative to Existing Frameworks. While the paper combines concepts appropriately, its originality is somewhat incremental. The noise-augmented LR conditioning interpolates between SR3 and InDI as acknowledged in Section 3.1. Similarly, the distillation loss in Eq. (9) is a constrained variant of forward distillation (Appendix B.2), and the core idea of aligning student and teacher ODE trajectories resembles BOOT. The contribution would be clearer if the authors better emphasized what properties uniquely arise from this specific combination in SR tasks.
>
>
> Our main contribution is **a principled distillation framework that enables a *continuous fidelity–realism trade-off* for single-step SR models**.
>
>
> ### **(1) Unifying SR3 and InDI is not merely incremental**
> While noise-augmented LR conditioning *recovers* SR3 and InDI as special cases, our framework is the **first to unify these models within a single, controllable formulation**.
> This unification is not a cosmetic interpolation:
>
> - We show experimentally (Table 7 in the updated version) that **intermediate values of the LR noise level $\sigma_p$** achieve a regime with **strong reconstruction quality and only a few sampling steps (straight trajectory)**, which neither SR3 nor InDI exhibit individually.
> - This intermediate regime is **a novel operational point** enabled by our formulation.
>
>
> ### **(2) Distillation objective: related to forward distillation, but fundamentally different**
> We agree that Eq. (9) resembles forward distillation, but the resemblance is superficial:
>
> - Our formulation was developed concurrently (see Appendix B.2), and serves as the discrete-time counterpart of the forward distillation loss.
> - Forward distillation requires continuous-time handling and Jacobian terms (e.g., MeanFlow), which is **computationally heavy**.
> - In contrast, our constrained distillation loss is **specifically designed for SR**, avoids Jacobians entirely (low distillation cost), and leads to a **practical, stable, and efficient** one-step model.
>
> Thus, even if they share conceptual ancestry, the objectives differ substantially in motivation, structure, and computational implications.
>
>
> ### **(3) Relation to BOOT: similarity in spirit, not in contribution**
> Our method aligns teacher and student trajectories, and this places us in the broad family of trajectory-matching distillation methods (e.g., BOOT, consistency models, MeanFlow). However:
>
> - BOOT focuses on text-to-image diffusion models and applies Signal-ODE matching. In commparison, our work is built for flow models and presents a much simpler and intuitive distillation loss.
> - Our work is **SR-specific**, built on rectified flow, and introduces **fidelity–realism controllability**, which BOOT does not address.
> - Trajectory matching is a **technical mechanism**, not the conceptual contribution.

---

> ### Author Response · Authors · 2025-11-21
> **Response to Reviewer cMY2 (2/3)**
>
> > [**W2 & Q1 & Q2**] [*W2*]Teacher dependence is acknowledged but under explored empirically. Section C mentions that the student's performance is constrained by the teacher's capability, but the experiments (Table 7) explore only a single ResShift teacher. The lack of systematic testing across multiple teachers limits understanding of the method's robustness when teacher quality varies or exhibits bias. [*Q1*] Clarification on Table 7: Could the authors explain the distinction between "OFTSR (ResShift teacher)" and "OFTSR (pre-train + distill)"? [*Q2*] Clarification on Table 8: Two OFTSR rows are listed with identical NFEs but differing parameter counts (118.6M+55.3M vs 552.8M). Do these correspond to different architectures (e.g., ResShift-based vs standalone OFTSR) or to inclusion/exclusion of VAE components?
>
> We appreciate the reviewer's observation regarding teacher dependence and have expanded the discussion in Section 4.1 to clarify the teacher models used in our study.
>
>
> ### **(1) Teacher dependence is inherent to all distillation frameworks**
> As with any teacher–student distillation setup, a student trained **solely** under teacher supervision cannot exceed the teacher's intrinsic capability.
> We explicitly acknowledge this limitation in Appendix Section C and emphasize that our method follows the same constraint.
>
>
> ### **(2) Additional clarification on teacher choices**
> We have added a dedicated paragraph in Section 4.1 describing the teacher models we use and referencing the results tables.
> To be specific, we employ:
> - Our pre-trained teacher models using Guided Diffusion backbone (pixel space, see Table 1, 2, 3), and ResShift backbone (latent space, see Table 4, 5, 6).
> - Additionally, we add new experiments with a stronger teacher (DIT4SR) in Table 6, demonstrating the scalability of our distillation framework.
> - In the original Table 7, 8 (now Table 9, 10), we use the pre-trained ResShift as teacher model to have fair comparison with prior arts and fair computation cost comparison.
>
> ### **(3) Clarification on Table 7 and Table 8 (now Table 9 and Table 10)**
>
> #### **(3.1) Clarification on Table 7 (now Table 9)**
> The two rows correspond to **different teacher sources**:
>
> - **OFTSR (pre-train + distill):** We first train a teacher using our *noise-augmented conditional flow* (Sec. 3.1), then distill it into a one-step model.
> - **OFTSR (ResShift teacher):** We directly use an **off-the-shelf ResShift checkpoint** as the teacher and apply our distillation algorithm.
>
> This distinction shows that OFTSR is **agnostic to the teacher architecture** and works effectively with both in-house and external teachers.
>
> #### **(3.2) Clarification on Table 8 (now Table 10)**
> The two OFTSR rows are the same configuration as in Table 7 (now Table 9), and they differ in **model architecture and whether a VAE is involved**:
>
> - **552.8M parameters (pre-train + distill):** Distilled from our **own trained teacher** using the guided diffusion backbone, which operates **without a VAE**. The larger parameter count reflects its standalone architecture.
> - **118.6M + 55.3M parameters (ResShift teacher):** Distilled from the **ResShift teacher**, which a *latent diffusion model*. The extra 55.3M corresponds to the ResShift VAE.
>
> We have updated the revised tables to make this correspondence clear.

---

> ### Author Response · Authors · 2025-11-21
> **Response to Reviewer cMY2 (3/3)**
>
> > [**W3**] Comparison with one step SR can be expanded. The experimental comparison omits some recent one-step SR approaches [1, 2], which could contextualize the claimed state-of-the-art results. Including these would strengthen the empirical claims.
>
>
> We thank the reviewer for pointing out the importance of comparing our distillation algorithm to recent one-step SR approaches.
> With the SD-based teacher model DIT4SR, we have expanded the comparison to include representative state-of-the-art one-step baselines, including **SinSR-1s, CTMSR-1s, AddSR-1s, OSEDiff-1s, and TSDSR-1s**, evaluated on both synthetic (DIV2K-Val) and real-world (RealLQ250) real SR benchmarks in the following updated tables:
>
> ## DIV2K-Val Dataset
>
> | Metric | SinSR-1s | CTMSR-1s | AddSR-1s | OSEDiff-1s | TSDSR-1s | **Ours (ResShift)-1s** | **Ours (DiT4SR)-1s** |
> |--------|----------|----------|----------|------------|----------|------------------------|---------------------|
> | PSNR ↑ | *24.50* | **24.87** | 22.39 | 23.86 | 22.17 | 23.91 | 22.80 |
> | SSIM ↑ | 0.6136 | **0.6349** | 0.5652 | *0.6233* | 0.5680 | 0.6073 | 0.5774 |
> | LPIPS ↓ | 0.3164 | 0.3011 | 0.3728 | 0.2896 | **0.2679** | 0.3226 | *0.2716* |
> | DISTS ↓ | 0.2110 | 0.2102 | 0.2387 | 0.1999 | *0.1901* | 0.2081 | **0.1889** |
> | FID ↓ | 131.96 | 126.49 | 133.78 | 100.53 | *103.49* | 133.30 | **98.27** |
> | NIQE ↓ | 6.1721 | 5.3036 | 5.9929 | 4.9741 | **4.6621** | 4.9061 | *4.8399* |
> | MUSIQ ↑ | 64.26 | 66.59 | 63.39 | 68.53 | **71.19** | 68.71 | *70.25* |
> | MANIQA ↑ | 0.5442 | 0.5146 | 0.5657 | *0.6111* | 0.6010 | 0.5464 | **0.6145** |
> | CLIPIQA ↑ | 0.6687 | 0.6602 | 0.5734 | 0.6692 | *0.7221* | 0.6545 | **0.7233** |
>
> ## RealLQ250 Dataset
>
> | Metric | SinSR-1s | CTMSR-1s | AddSR-1s | OSEDiff-1s | TSDSR-1s | **Ours (ResShift)-1s** | **Ours (DiT4SR)-1s** |
> |--------|----------|----------|----------|------------|----------|------------------------|---------------------|
> | NIQE ↓ | 5.8200 | 4.5835 | 4.9235 | 3.9656 | **3.4868** | 4.0731 | *3.7802* |
> | MUSIQ ↑ | 63.73 | 68.00 | 66.82 | 69.55 | *72.09* | 67.32 | **72.60** |
> | MANIQA ↑ | 0.5161 | 0.5078 | 0.5304 | 0.5782 | 0.5829 | 0.5287 | **0.5904** |
> | CLIPIQA ↑ | 0.6990 | 0.6706 | 0.6437 | 0.6725 | *0.7221* | 0.6532 | **0.7252** |
> | LIQE ↑ | 3.2578 | 3.3373 | 3.4929 | 3.9039 | *4.0834* | 3.7211 | **4.1122** |
>
> **Note:** ↑ indicates higher is better, ↓ indicates lower is better. **Bold** indicates best results, *italic* indicates second best.
>
> > Duan, Zheng-Peng, Jiawei Zhang, Xin Jin, Ziheng Zhang, Zheng Xiong, Dongqing Zou, Jimmy S. Ren, Chunle Guo, and Chongyi Li. "Dit4sr: Taming diffusion transformer for real-world image super-resolution." In Proceedings of the IEEE/CVF International Conference on Computer Vision, pp. 18948-18958. 2025.
>
> > [**W4**] Editorial issues. Minor editorial redundancies exist, such as repeated inference equations 13-14.
>
> We thank the reviewer for pointing out the editorial redundancies.
> We have carefully revised the manuscript to eliminate such redundancies.

---

### Author Response · Authors · 2025-11-30
**General Response**

Dear AC, SAC, and Reviewers,

We sincerely thank you for your time and constructive feedback throughout the review process.
The comments have significantly strengthened the paper and guided us toward improved clarity and performance.

We appreciate that the manuscript is regarded as well-written, empirically solid, and clearly motivated.
Below we summarize our main clarifications and improvements.

## **1. Novelty: A principled one-step SR framework with continuous fidelity-realism control**

Several reviewers questioned whether the method is incremental relative to SR3, InDI, BOOT, or forward distillation.
Our main contribution is **not** a minor refinement of distillation methods - it is a **principled framework that transfers the ability of controlling *fidelity-realism trade-off* from the multistep teacher models into a *single-step* SR generator**, which has not been demonstrated before.

- Our **noise-augmented conditional flow** is the *first* to unify SR3 and InDI **non-trivially**: intermediate $\sigma_p$ values yield a **new operating regime** balancing sampling steps and realism, which neither SR3 nor InDI achieve individually.
- Our **distillation algorithm** is a **discrete counterpart** of forward distillation, specifically designed for SR and avoiding Jacobians calculation, making it **efficient, stable, and practical**.
- Although conceptually related to BOOT/consistency models, we provide a clean and clear derivation from the flow perspective and more importantly, achieves **controllable fidelity-realism trade-offs** in the one-step regime.

The novelty lies in the **new capability** enabled by the principled combination of these components.

---

## **2. Expanded experiments: SOTA baselines, stronger teachers, and real-world validation**

In response to reviewers, we substantially expanded our experiments.

### **(1) Added comparisons with recent one-step SR models**

Table 6 now include comparison to **SinSR-1s, CTMSR-1s, AddSR-1s, OSEDiff-1s, TSDSR-1s**, covering the latest state-of-the-art.

### **(2) Distillation from stronger and diverse teachers**

We demonstrate teacher-agnostic scalability by distilling from:

- **Self-trained teacher with Guided Diffusion** backbone(pixel-space)
- **Self-trained teacher with ResShift** backbone(latent-space)
- **Off-the-shelf ResShift** teacher (latent-space)
- **DiT4SR (SD3.5-based)** high-capacity SR models

With DiT4SR teacher, our one-step model reaches competitive or superior realism metrics (NIQE, MUSIQ, MANIQA, CLIPIQA).

### **(3) Real-world robustness**

We newly evaluate on RealSet80 and RealLQ250, containing real degraded images (Tables 5 and 6).
Figure 16 provides additional qualitative robustness under domain shift.

### **(4) Comparison with deterministic SR (e.g., SwinIR, NAFNet)**

We compare against SwinIR and NAFNet, showing that our one-step model provides far better perceptual realism while maintaining a similar inference cost (one forward pass).

---

## **3. Resolution generalization: Strong high-resolution performance without retraining**

Although our main model is trained at **64→256**, it generalizes remarkably well:

- Works directly on **128→512** for RealLQ250.
- Can handle **arbitrary resolutions** (Fig. 17).
- Performs strongly on **RealSet80**, where native LR/HR sizes vary widely.
- When using DiT4SR teachers, the distilled model supports native **512x512** SR.

Thus, OFTSR inherits the teacher’s scalability and remains robust across resolutions.

---

## **4. Stability and interpretability of the trade-off parameter *t***

Across all datasets and teacher models:

- The realism-fidelity trade-off is **smooth and stable** for all $ t \in [0,1] $.
- No artifacts appear at extreme values $ t=0 $ or $ t=1 $ (Fig. 13).
- Out-of-domain values (e.g., $ t=2 $, $ t=-0.5 $) degrade as expected (Appendix Fig. 10).
- If a target metric is specified, *t* can be **automatically selected** via validation optimization.

---

## **Closing Remarks**

We appreciate the reviewers' comprehensive comments.
With clearer explanations, expanded experiments, and additional analyses, we hope it is now evident that:

**Our contribution is a principled framework enabling a continuous fidelity-realism trade-off within a *single-step* SR model - a novel capability not offered by prior distillation or consistency-based approaches.**

We have updated the manuscript to reflect these clarifications and improvements.

Sincerely,

The Authors

---

### Meta-Review · Area_Chair_a7D8 · 2026-01-07

**Summary:**

The paper introduces a conditional rectified flow framework for one-step image super-resolution that enables continuous control between fidelity and realism. The proposed approach offers flexible, single-step inference and demonstrates strong quantitative and qualitative performance across diverse datasets, outperforming or matching baselines while delivering significant efficiency gains. The main concerns raised by reviewers focused on the originality of the ideas and the empirical validation of the method. In their rebuttal, the authors addressed all reviewer concerns, providing convincing responses on positioning of the paper as compared to other related works, and additional empirical evidence supporting the benefits of their approach. I believe this paper makes a valuable contribution to the field, and I recommend its acceptance.

**Reviewer Concerns:**

A concern raised by the reviewers was the incremental novelty of the paper. The authors thoroughly addressed this by positioning their work as a principled framework that enables control over the fidelity–realism trade-off compared to existing approaches. In addition, they significantly strengthened the empirical validation by providing additional experiments, including varied settings and more baselines. Overall, all reviewer comments were thoroughly addressed in the authors’ rebuttal.

**Reviewer Scores:**

Reviewers cMY2 and U3BC initially gave scores of 4, but since all their concerns were addressed, it is likely they would have raised their scores. Reviewer 835P gave a high score of 8 and would likely have maintained or possibly increased it. Reviewer dEo3 started with a score of 4 and expressed an intention to raise it to 6 after the rebuttal. Finally, reviewer vEMX gave a score of 6 and would likely have either kept it or increased it to 8 following the rebuttal.

---

### Decision · Program_Chairs · 2026-01-26

Accept (Poster)